# Rewarding the Rare: Maverick-Aware Shapley Valuation in Federated Learning

**Mengwei Yang**                                                                 *mengwey@uci.edu*
*Department of Electrical Engineering and Computer Science*
*University of California, Irvine*

**Baturalp Buyukates**                                                      *b.buyukates@bham.ac.uk*
*School of Computer Science*
*University of Birmingham*

**Athina Markopoulou**                                                              *athina@uci.edu*
*Department of Electrical Engineering and Computer Science*
*University of California, Irvine*

**Reviewed on OpenReview:** *https://openreview.net/forum?id=JtybGfTUdq*

## Abstract

Federated Learning (FL) allows clients to train a model collaboratively without sharing their private data. Shapley value (SV) provides a principled way to quantify client contributions in FL. However, existing SV methods use uniform per-class weighting during validation, treating all classes as equally important. This uniform weighting breaks down in the presence of clients with underrepresented or rare classes, also referred to as *Mavericks*. Such clients are often undervalued due to lower model performance on these challenging classes, despite their critical role in improving generalization. To address this, we introduce a Maverick-aware Shapley valuation framework that reweights validation scores based on per-class accuracy, assigning greater importance to classes where models perform poorly. Building on this, we design FedMS, a **M**averick-**S**hapley client selection mechanism that leverages our refined contribution scores to guide intelligent client selection. Experiments on benchmark datasets demonstrate that FedMS improves model performance and better recognizes valuable client contributions, even under scenarios involving adversaries, free-riders, and skewed or rare-class distributions.

## 1 Introduction

Federated Learning (FL) enables collaborative model training across distributed clients without requiring direct access to raw data (McMahan et al., 2017). Clients communicate only local model updates with a central server, keeping their data on-device and thereby enhancing privacy. While this distributed setup offers privacy advantages, it also introduces significant challenges in understanding and leveraging the role of each client in the learning process.

A key challenge in FL is how to evaluate client contributions accurately. Shapley value (SV) has become a widely adopted tool for evaluating client contributions based on their marginal impact on model performance. In practice, validation accuracy is commonly used as the utility function for SV estimation, as it provides an effective and interpretable measure of model quality across clients and rounds. However, SV estimation in FL typically aggregates validation accuracy uniformly across all classes, implicitly assuming that each class is equally important, equally frequent, and equally easy to learn. Such an assumption fails in settings where data is imbalanced, particularly in the presence of clients holding rare or underrepresented classes.

These clients, referred to as Mavericks (Huang et al., 2022), hold rare data in FL, typically consisting of one or more classes that they exclusively own and that are absent from other clients. Mavericks play a vital role in improving model generalization. Incorporating such clients helps mitigate algorithmic bias, leading to more robust and trustworthy machine learning systems. For example, when training an FL model for a disease classification task (Song et al., 2024; Chen et al., 2023), most hospitals (*i.e.,* clients) can possess data indicating common diseases such as flu or other frequent infections. However, few hospitals may possess more rare disease datasets such as for leukemia or thyroid cancers, making them Mavericks for this learning task. Another example of Mavericks is people with rare accents for training voice-activated AI systems like Amazon's Alexa and Google's Home Assistant. While the majority of these devices contain native accent data, a few of them contain data from users with non-native accents. Recent studies (Kamegne et al., 2025; Michel et al., 2025) report that these voice systems struggle to understand non-native accents. This performance disparity indicates a biased performance and demonstrates the importance of training with rare (or less common) data from Mavericks, to create models that "speak" to everyone.

Despite their importance, Mavericks are systematically undervalued by existing SV-based methods. Rare classes are often harder to learn and tend to yield lower validation accuracy, causing clients that contribute such data to appear less useful when uniform class weighting is applied during evaluation. As a result, SV assigns disproportionately low contribution scores to Mavericks. This misvaluation can lead to biased global models, suboptimal training dynamics, and disincentivized participation from Mavericks whose data is both rare and valuable.

In this paper, we propose a principled reweighting of the validation score used in SV estimation, incorporating class difficulty to enable more accurate and equitable assessment of client contributions. Specifically, we introduce a class-aware validation strategy that places greater emphasis on performance over rare and hard-to-learn classes. This reweighting mitigates the bias introduced by uniform validation and more faithfully captures the true utility of each client, particularly Mavericks, during training.

Building on this reweighted valuation, we introduce FedMS, a Maverick-aware client selection mechanism that leverages the refined Shapley scores to guide participation in each round. In contrast to prior selection strategies, FedMS adaptively prioritizes clients based on the difficulty and rarity of the classes they contribute, enabling more effective utilization of Mavericks and most valuable participants. This leads to improved performance and robustness, particularly in the presence of Mavericks in FL.

In this paper, we offer a principled approach for valuing and leveraging Mavericks who contribute rare or hard-to-learn classes in FL. Our key contributions are as follows:

- We identify a key limitation in existing SV-based methods, showing that uniformly weighted validation scores systematically undervalue clients with rare or underrepresented classes, thus limiting both contribution estimation and model generalization.

- We propose Maverick-Shapley, a Maverick-aware Shapley valuation mechanism that reweights per-class validation scores based on empirical class difficulty, leading to more accurate and robust contribution assessments.

- We develop FedMS, a contribution-guided client selection strategy that leverages Maverick-Shapley to prioritize and effectively utilize valuable clients, including Mavericks, even in adversarial and free-rider settings.

## 2 Related Work

**Contribution Evaluation via Gradient Shapley Methods.** SV, a classic tool from cooperative game theory, has been widely adopted in FL to quantify client contributions based on their marginal impact on model performance. However, exact SV computation is infeasible in FL due to the need to retrain models over all possible client permutations. To address this, a line of work has focused on gradient-based SV approximations that avoid costly retraining by leveraging clients' gradient updates. Reference (Song et al., 2019) introduces two SV estimators: One-Round (OR), which computes SV post-training, and Multi-Round

(MR), which updates SV estimates throughout training. Truncated Multi-Rounds Construction (TMR) (Wei et al., 2020) improves efficiency by truncating unnecessary sub-model evaluations with a decay factor. Guided Truncation Gradient Shapley (GTG) (Liu et al., 2022) further refines this process by combining guided Monte Carlo sampling with gradient-based truncation. Despite these advances, existing SV approximations such as OR, MR, TMR, and GTG commonly rely on uniformly weighted validation accuracy as the utility function, making them ineffective in the presence of rare or underrepresented classes. Prior work (Huang et al., 2022; Buyukates et al., 2023) has shown that such methods systematically undervalue Mavericks, as their lower accuracy on rare classes leads to reduced contribution scores under uniform weighting. In this work, we address this fundamental limitation by proposing a Maverick-aware SV framework that reweights per-class validation scores according to class difficulty, enabling more accurate valuation of Mavericks. Although both CS-Shapley (Schoch et al., 2022) and ShapFed (Tastan et al., 2024) incorporate class-wise SV, their goals differ from ours. CS-Shapley decomposes instance contributions into in-class and out-of-class accuracy for attribution in centralized ML, while ShapFed uses class-wise SV to construct a weighted aggregation scheme in FL. In contrast, our FedMS explicitly targets Mavericks with rare data distributions, recognizing their rare class contributions to enhance client selection in FL, which CS-Shapley and ShapFed has not considered.

**Client Selection in the Presence of Mavericks.** In FL, Mavericks are typically underrepresented in the broader client population. Although they are crucial for improving model generalization, these clients are often overlooked or underutilized in standard FL training. The FedAvg algorithm (McMahan et al., 2017), which randomly samples clients in each round, does not prioritize those contributing rare or critical information (Fu et al., 2023). Several client selection strategies have been developed to address data heterogeneity. S-FedAvg (Nagalapatti & Narayanam, 2021) combines FedAvg with SV estimation to favor clients with higher estimated contributions. GreedyFed (Singhal et al., 2024) selects clients using a fast SV approximation method known as GTG (Liu et al., 2022). Power of Choice (PoC) (Cho et al., 2022) schedules clients with the highest local loss in each round. However, none of these methods explicitly identifies or prioritizes Mavericks, nor do they effectively capture the contribution of clients with rare data. FedEMD (Huang et al., 2022) is one of the few methods that explicitly increases the selection probability of Mavericks, using the Wasserstein distance between local and global class distributions. While this improves their sampling frequency, it does not provide a principled mechanism to quantify their actual contributions during training. On the other hand, we develop a Maverick-aware Shapley client valuation mechanism and leverage it to guide client selection in each training round in the presence of clients with rare data. By prioritizing clients based on the rarity and difficulty of the classes they contribute, our method enables more effective utilization of Mavericks across diverse data settings.

**Data Heterogeneity in FL.** FL suffers from performance degradation under heterogeneous client data. To address this, numerous works have studied different types of heterogeneity, including covariate shift (where the feature distribution varies) and concept shift (where class distributions differ). For instance, FedBN (Li et al., 2020b) employs local batch normalization to alleviate feature shift, while IFCA (Ghosh et al., 2020) and FedDrift (Jothimurugesan et al., 2023) partition clients into clusters to address concept shift. Personalized FL approaches such as FedALA (Zhang et al., 2023) and meta-learning methods like (Fallah et al., 2020) aim to adapt models to diverse client objectives. Our setting, referred to as the Maverick scenario, can be seen as an extreme case of concept shift, where certain clients exclusively hold one or more classes that are entirely absent from the data of other clients. This creates sharp class-wise imbalances that challenge the standard evaluation and aggregation strategies in FL.

## 3 Problem Setup and Background

In this section, we first formalize the FL framework (McMahan et al., 2017) and define Mavericks (Huang et al., 2022). We then give an overview of the SV-based methods for evaluating the contribution of clients. Figure 1 illustrates the data distribution among clients across three Mavericks scenarios we consider in this work.

**Federated Learning (FL).** We consider a standard FL setup consisting of a central server and a set of clients $\mathcal{K} = \{1, 2, \ldots, I\}$. Each client $i \in \mathcal{K}$ holds a local dataset $\mathcal{D}_i$ of size $n_i = |\mathcal{D}_i|$. The total number

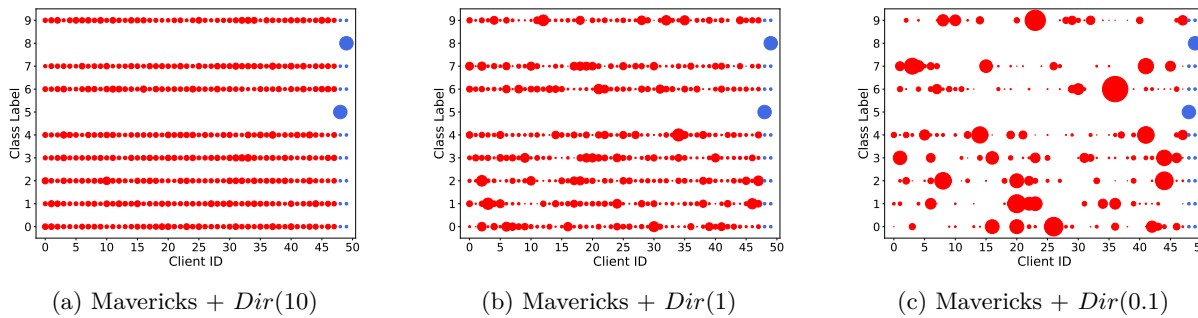

(a) Mavericks + $Dir(10)$     (b) Mavericks + $Dir(1)$     (c) Mavericks + $Dir(0.1)$

Figure 1: Data sample distribution across clients in three Mavericks scenarios for MNIST and CIFAR-10 datasets. Mavericks are shown as blue circles, and non-Mavericks as red. Circle size indicates sample count, with heterogeneity controlled by $\gamma$ in $Dir(\gamma)$. Higher $\gamma$ means lower heterogeneity. (a) is the most IID setting with identical distributions for all but Mavericks. (b) shows moderate heterogeneity, and (c) is the most Non-IID. In all cases, the Mavericks exclusively hold certain class labels (e.g., class 5 and class 8), while still containing samples from other classes.

of samples across all clients is $n = \sum_{i \in \mathcal{K}} n_i$. Each data point is a pair $(\boldsymbol{x}, y)$, where $\boldsymbol{x}$ denotes the feature vector and $y \in \mathcal{M} = \{1, 2, \ldots, C\}$ is the corresponding class label.

Let $\boldsymbol{w}$ denote the parameters of the global model. The objective of FL is to minimize the weighted average of local losses:

$$\min_{\boldsymbol{w}} \mathcal{L}(\boldsymbol{w}) = \sum_{i \in \mathcal{K}} \frac{n_i}{n} \mathcal{L}_i(\boldsymbol{w}), \tag{1}$$

where the local loss at client $i$ is defined as:

$$\mathcal{L}_i(\boldsymbol{w}) = \frac{1}{n_i} \sum_{(\boldsymbol{x}, y) \in \mathcal{D}_i} \ell(\boldsymbol{w}; \boldsymbol{x}, y), \tag{2}$$

and $\ell(\boldsymbol{w}; \boldsymbol{x}, y)$ denotes the per-sample loss.

The FL training proceeds in communication rounds and involves the following steps: **(i) Initialization:** The server initializes the global model parameters $\boldsymbol{w}^0$ and broadcasts them to all clients. **(ii) Client Selection:** In round $t$, the server selects a subset of clients $\mathcal{K}^t \subseteq \mathcal{K}$, with $|\mathcal{K}^t| = I^t$, based on a client selection strategy $\pi$. **(iii) Local Update and Aggregation:** Each selected client $i \in \mathcal{K}^t$ performs local training and returns an updated model $\boldsymbol{w}_i^t$ to the server. The server then aggregates these updates to form the new global model:

$$\boldsymbol{w}^{t+1} = \sum_{i \in \mathcal{K}^t} \frac{n_i^t}{\sum_{j \in \mathcal{K}^t} n_j^t} \boldsymbol{w}_i^t. \tag{3}$$

Steps (ii) and (iii) are repeated until convergence of the global model.

**Mavericks.** The term Maverick was introduced by Huang et al. (2022) to refer to clients that exclusively hold data from one or more class labels; that is, no other client in the FL system possesses samples from those classes. Let $\mathcal{M}_{\text{mav}} \subseteq \mathcal{M}$ denote the set of class labels exclusively owned by Mavericks. For a given client $i$ and class label $c \in \mathcal{M}$, we define $\mathcal{D}_i^c = \{(x, y) \in \mathcal{D}_i \mid y = c\}$ as the subset of data with label $c$.

A client $i$ is a Maverick if $\mathcal{D}_i$ includes samples from one or more classes in $\mathcal{M}_{\text{mav}}$, i.e.,

$$\mathcal{D}_i = \bigcup_{c \in \mathcal{M}_{\text{mav}}} \mathcal{D}_i^c \cup \bigcup_{c \notin \mathcal{M}_{\text{mav}}} \mathcal{D}_i^c. \tag{4}$$

A non-Maverick client only contains data from the remaining classes:

$$\mathcal{D}_i = \bigcup_{c \notin \mathcal{M}_{\text{mav}}} \mathcal{D}_i^c. \tag{5}$$

In practice, multiple clients may jointly own rare classes; we refer to these as shared Mavericks.

**Shapley Value (SV) for Client Valuation in FL.** SV (Shapley, 1971; Ghorbani & Zou, 2019) of client $i$ is given by

$$\phi_i(\mathcal{K}, \mathcal{V}) = \frac{1}{|\mathcal{K}|} \sum_{\mathcal{Q} \subseteq \mathcal{K} \setminus \{i\}} \frac{\mathcal{V}(\mathcal{Q} \cup \{i\}) - \mathcal{V}(\mathcal{Q})}{\binom{|\mathcal{K}|-1}{|\mathcal{Q}|}}, \tag{6}$$

where $\phi_i$ denotes the SV for client $i$, and $\mathcal{Q}$ represents a subset of the client set $\mathcal{K}$. The utility function $\mathcal{V}(\cdot)$ can take any form that quantifies the value of a given subset; in this work, as in much of the FL literature, we use validation accuracy to measure utility.

In cooperative game theory, the SV framework is utilized to calculate the contribution of each player in a coalition. Computing the exact SV in Equation (6) requires retraining the FL model for all possible subsets of clients, which is computationally prohibitive. To overcome this limitation, we adopt SV approximation methods commonly used in the literature. Crucially, our proposed Maverick-Shapley mechanism augments existing SV approximations, including MR (Song et al., 2019), TMR (Wei et al., 2020), and GTG (Liu et al., 2022), to enable Maverick-aware estimation of client contributions, which we discuss in the next section.

## 4 Proposed Method: FedMS

In this section, we first introduce the Maverick-Shapley contribution score, a class-wise SV mechanism for accurately estimating client contributions. The detailed steps of Maverick-Shapley are presented in Algorithm 1. We then integrate this valuation into a client selection mechanism, termed FedMS, which prioritizes the most valuable clients in each round based on their Maverick-Shapley contribution scores (see Algorithm 2). Table 6 in Appendix presents the main parameters and notations of our proposed method.

### 4.1 Maverick-Shapley

This section describes the proposed Maverick-Shapley client contribution score. When training a model for multi-class tasks, the difficulty of learning each class is different. Particularly in the presence of Mavericks, rare classes are harder to learn than the others. In order to differentiate between classes and accurately compute the contribution of each client (Mavericks and non-Mavericks alike), we propose a class-wise SV-based contribution score. In particular, we use the class-wise accuracy[1] as the utility function in SV computations and aggregate class-wise contributions into a single number using the difficulty level of each class. Class-wise accuracy is calculated as

$$\mathcal{V}_{class}^c(\boldsymbol{w}; \mathcal{D}_{\text{val}}) = \frac{N^{cc}}{\sum\limits_{j \in \mathcal{M}} N^{cj}}, \quad \forall c \in \mathcal{M}, \tag{7}$$

where $w$ is a given model, $\mathcal{D}_{\text{val}}$ is validation dataset at the server, $N^{cj}$ represents the number of validation data points of class $c$ predicted as class $j$, $\mathcal{M}$ is set of class labels. Our main distinction in SV computation is the fact that we compute it in a class-wise manner to better capture the diverse resources of Mavericks.

In each FL round, after receiving model updates from the participating clients, the server computes the SV of a client $i$ for class $c$, $\phi_i^c$, by utilizing a gradient-based SV approximation method of its choice using (7). It then computes the accumulated class-wise SVs $S_i^c$ using a decay factor $\alpha$ as

$$S_i^c = \alpha * S_i^c + (1 - \alpha) * \phi_i^c, \quad \forall i \in \mathcal{K}^t, \forall c \in \mathcal{M}. \tag{8}$$

---

[1]The class-wise accuracy is equivalent to the per-class recall, measuring the proportion of correctly predicted samples within each class.

Finally, the server computes the Maverick-Shapley contribution score of each client $i$ as a weighted sum of $S_i^c$s in each training round[2]

$$\hat{S}_i = \sum_{c \in \mathcal{M}} \beta^c \cdot S_i^c, \quad \forall i \in \mathcal{K}, \tag{9}$$

where $\beta^c$ denotes the class difficulty of class $c$ in the current training round, adaptively adjusting the impact of each class in the contribution scores such that

$$\beta^c = \frac{exp\left(\frac{1-\mathcal{V}_{class}^c(\boldsymbol{w};\mathcal{D}_{\text{val}})}{T}\right)}{\sum\limits_{c \in \mathcal{M}} exp\left(\frac{1-\mathcal{V}_{class}^c(\boldsymbol{w};\mathcal{D}_{\text{val}})}{T}\right)}, \quad \forall c \in \mathcal{M}, \tag{10}$$

where the temperature $T$ controls the distribution. Since the difficulty of learning each class is dynamically changing, the server updates the class difficulty $\boldsymbol{\beta}$ and the contribution score $\hat{S}$ in each round.

Our proposed Maverick-Shapley approach is universally applicable to the existing gradient-based SV approximations, as we simply change the utility function and utilize class difficulties $\boldsymbol{\beta}$. In this paper, we integrate Maverick-Shapley approach with three gradient-based SV approximations including MR (Song et al., 2019), TMR (Wei et al., 2020) and GTG (Liu et al., 2022). In the ensuing, we describe Maverick-Shapley GTG. Details on Maverick-Shapley MR and Maverick-Shapley TMR are given in the Appendix.

**Maverick-Shapley GTG (MS-GTG).** MS-GTG is built on top of the Guided Truncation Gradient Shapley (GTG) approach (Liu et al., 2022) by modifying the utility function and incorporating class difficulties as shown in Algorithm 1. The GTG approach designs the guided sampling (Line 11 in Algorithm 1) to improve the efficiency of subset model reconstructions and removes unnecessary reconstructions using truncation techniques (Line 7 and 15 in Algorithm 1). We utilize the following convergence criterion (Liu et al., 2022) for MS-GTG

$$\frac{1}{|\mathcal{K}^t| \times m} \sum_{j=1}^m \sum_{i=1}^{|\mathcal{K}^t|} \frac{|\phi_i^{[r]} - \phi_i^{[r-j]}|}{|\phi_i^{[r]}|} < 0.05, \tag{11}$$

where $r$ is the number of iterations inside MS-GTG (Line 10 in Algorithm 1), $\phi_i^{[r]}$ is a weighted sum of its class-wise SV, $\phi_i^{[r]} = \sum\limits_{c \in \mathcal{M}} \beta^c \cdot \phi_i^c$, and $m$ represents the number of preceding terms considered in the sum.

After guided sampling, the server computes the class-wise SV (line 22 in Algorithm 1), identifies the coreset $\hat{\mathcal{K}}^t$ and updates the class difficulty $\boldsymbol{\beta}$ (lines 24 and 26 in Algorithm 1). The coreset is defined as the subset of sampled clients $\mathcal{K}^t$ whose aggregation maximizes total class-wise validation accuracy. The algorithm thus outputs the per-client SV $\boldsymbol{\phi}$, the class difficulty $\boldsymbol{\beta}$, and the coreset $\hat{\mathcal{K}}^t$.

## 4.2 Maverick-Shapley Client Selection

The Maverick-Shapley contribution score we propose is a principled way for assessing the value of clients with rare data in FL. In this section, we describe how these contribution scores can be utilized by the server to make client selection in each round. In a nutshell, based on these contribution scores, the server selects the most contributing clients in each FL training round. This process is described in Algorithm 2. In each round the server calculates the selection probability of clients according to their contribution scores $\hat{S}$ as

$$P_{\hat{S},i} = \frac{exp(\hat{S}_i)}{\sum\limits_{i \in \mathcal{K}} exp(\hat{S}_i)}, \quad \forall i \in \mathcal{K}, \tag{12}$$

and samples clients based on $P_{\hat{S}}$ in each round. Since Mavericks exclusively owns certain classes, they are the primary contributors for these rare classes and are more likely to be selected when the model underperforms on them.

---

[2]In the first round, the server queries model updates from all clients and initializes contribution scores based on the class-wise accuracy of these updates.

---

**Algorithm 1:** Maverick-Shapley GTG (MS-GTG)

---

**1** **Input**: Updated client models $\{\boldsymbol{w}_i^t\}_{i \in \mathcal{K}^t}$; current server model $\boldsymbol{w}^t$; validation dataset at server $D_{val}$; class-wise accuracy function $\mathcal{V}_{class}(\cdot)$; $\mathcal{M}$: set of class labels; $\mathcal{K}^t$: set of clients in round $t$;.

**2** **Hyperparameters:** Error threshold $\epsilon_b$, $\epsilon_i$, temperature T.

**3** **Initialize:** $\phi_i = 0, \forall i \in \mathcal{K}^t, r = 0$

**4** Compute $\boldsymbol{w}^{t+1} = \texttt{ModelAverage}(n_i, \{\boldsymbol{w}_i^t\}_{i \in \mathcal{K}^t})$

**5** $v_0 = \mathcal{V}_{class}(\boldsymbol{w}^t; \mathcal{D}_{\text{val}}), v_N = \mathcal{V}_{class}(\boldsymbol{w}^{t+1}; \mathcal{D}_{\text{val}}),$

**6** *// between round truncation*

**7** **if** $|v_N - v_0| > \epsilon_b$ **then**

**8**   **while** Convergence criteria not met **do**

**9**     **for** *client* $i \in \mathcal{K}^t$ **do**

**10**       $r = r + 1$

**11**       $\pi^{[r]} \leftarrow$ Random permutation of $\mathcal{K}^t \setminus \{i\}$ with $\pi^{[r]}[0] = i$

**12**       $v_0^{[r]} = v_0$

**13**       *// within-round truncation*

**14**       **for** $j = 1, \dots, n$ **do**

**15**         **if** $|v_N - v_{j-1}^{[r]}| \geq \epsilon_i$

**16**           $H = \pi^{[r]}[:j]$

**17**           $\widetilde{\boldsymbol{w}}_H = \texttt{ModelAverage}(\{\boldsymbol{w}_i^t\}_{i \in H}, \boldsymbol{w}^t)$

**18**           $v_j^{[r]} = \mathcal{V}_{class}(\widetilde{\boldsymbol{w}}_H; \mathcal{D}_{\text{val}})$

**19**         **else**

**20**           $v_j^{[r]} = v_{j-1}^{[r]}$

**21**         **for** *class* $c \in \mathcal{M}$ **do**

**22**           $\phi_{\pi^{[r]}[j]}^c = \frac{r-1}{r} \phi_{\pi^{[r]}[j]}^c + \frac{(v_j^{[r],c} - v_{j-1}^{[r],c})}{r}$

**23** *// Find coreset $\hat{\mathcal{K}}^t$ and its class-wise accuracy $\hat{v}$*

**24** $\hat{\mathcal{K}}^t, \hat{v} \leftarrow \text{argmax}_H \sum_{c \in \mathcal{M}} \mathcal{V}_{class}^c(\widetilde{\boldsymbol{w}}_H; \mathcal{D}_{\text{val}})$

**25** *// Obtain class difficulty $\boldsymbol{\beta}$*

**26** $\beta^c = \frac{exp(\frac{1-\hat{v}^c}{T})}{\sum_{c \in M} exp(\frac{1-\hat{v}^c}{T})}, \forall c \in \mathcal{M}$

**27** **return** $\boldsymbol{\phi}, \boldsymbol{\beta}, \hat{\mathcal{K}}^t$

---

Although clients are sampled from $P_{\hat{S},i}$, including all sampled clients $\mathcal{K}^t$ in aggregation is not always optimal. The collective contribution of a group depends on how their updates interact: some sampled clients may contribute redundant gradients or even destabilize the aggregation, while others (e.g., Mavericks with rare classes) provide complementary information that significantly improves generalization. To ensure aggregation emphasizes the most beneficial interactions, the server identifies the coreset $\hat{\mathcal{K}}^t$ through Maverick-Shapley and aggregates only their updates.

**Shapley Rewards (SR).** In each round, the server computes the class-wise SV of a selected client $i$ for each class $c$, $\phi_i^c$. It then calculates the SR of each client $i$ for round $t$ as a weighted sum of its class-wise SVs using the current class difficulty $\boldsymbol{\beta}$

$$R_i^t = \sum_{c \in \mathcal{M}} \beta^c \cdot \phi_i^c, \quad \forall i \in \mathcal{K}^t. \tag{13}$$

## 5 Evaluation

We comprehensively evaluate the effectiveness of our algorithm, FedMS, on two datasets against six baselines, in terms of (I) model performance, (II) contribution allocation across Mavericks and non-Mavericks, and (III) robustness against adversaries and free-riders.

---

**Algorithm 2:** FedMS: a Maverick-Shapley Client Selection Mechanism for FL

---

**1** **Input:** $T$: number of training rounds; $E$: number of local epochs; $\mathcal{K}^t$: set of clients in round $t$; $\mathcal{D}_i$: dataset of client $i$; $B$: minibatch size; $n_i^t$: dataset size of the $i$th client in round $t$; $\mathcal{M}$: set of class labels; $\mathcal{D}_{val}$: validation dataset; $\mathcal{V}_{class}(\cdot)$: class-wise accuracy function; $\eta_i$: learning rate at client $i$.

**2** **Server executes:**

**3**     Initialize $\boldsymbol{w}^0$, $\boldsymbol{\beta}$, $\hat{S}$

**4**     **for** *each round $t = 0, \dots T-1$* **do**

**5**         *// Sample clients from $P_{\hat{S},i}$.*

**6**         $P_{\hat{S},i} = \frac{exp(\hat{S}_i)}{\sum\limits_{i \in \mathcal{K}} exp(\hat{S}_i)}, \forall i \in \mathcal{K}$

**7**         $\mathcal{K}^t \leftarrow$ sample clients $\sim P_{\hat{S},i}$

**8**         **for** *each client $i \in \mathcal{K}^t$ in parallel* **do**

**9**             $\boldsymbol{w}_i^t \leftarrow$ UserUpdate $(\boldsymbol{w}^t, i)$

**10**         *// Calculate class-wise SV $\phi_i$, class difficulty $\boldsymbol{\beta}$ and coreset $\hat{\mathcal{K}}^t$.*

**11**         $\boldsymbol{\phi}, \boldsymbol{\beta}, \hat{\mathcal{K}}^t \leftarrow$ Maverick-Shapley $(\{\boldsymbol{w}_i^t\}_{i \in \mathcal{K}^t}, \boldsymbol{w}^t, D_{val}, \mathcal{V}_{class}(\cdot), \mathcal{M})$

**12**         *// Compute the accumulated class-wise SV $S_i$.*

**13**         $S_i^c = \alpha \cdot S_i^c + (1-\alpha) \cdot \phi_i^c, \forall i \in \mathcal{K}^t, \forall c \in \mathcal{M}$

**14**         *// Compute contribution score $\hat{S}_i$.*

**15**         $\hat{S}_i = \sum\limits_{c \in \mathcal{M}} \beta^c \cdot S_i^c, \forall i \in \mathcal{K}$

**16**         $\boldsymbol{w}^{t+1} \leftarrow \sum\limits_{i \in \hat{\mathcal{K}}^t} \frac{n_i^t}{\sum\limits_{i \in \hat{\mathcal{K}}^t} n_i^t} \boldsymbol{w}_i^t;$

**17** **function** UserUpdate $(\boldsymbol{w}^t, i)$:

**18**     **for** *each local epoch $e = 1 \dots E$* **do**

**19**         $\mathcal{D}_i^B \leftarrow$ select a minibatch of size $B \subseteq \mathcal{D}_i$

**20**         $\boldsymbol{w}_i^t \leftarrow \boldsymbol{w}_i^t - \eta_i \nabla \mathcal{L}_i(\mathcal{D}_i^B, \boldsymbol{w}_i^t)$

**21**     **return** $\boldsymbol{w}_i^t$ *to server*

---

## 5.1 Evaluation Setup

**Datasets.** We use five benchmark datasets, (1) MNIST (Deng, 2012) consisting of handwritten digits, with 60,000 samples for training and 10,000 for testing, and (2) CIFAR-10 (Krizhevsky et al., 2009) consisting of colored images of 10 classes, with 50,000 samples for training and 10,000 for testing. Another 3 datasets are from MedMNIST (Yang et al., 2021; 2023), a large-scale MNIST-like collection of standardized biomedical images. (3) BloodMNIST (Acevedo et al., 2020) including blood cell microscope data, with 11,959 samples for training, 1,712 samples for validation and 3,421 for testing. (4) OrganAMNIST (Bilic et al., 2023; Xu et al., 2019) including abdominal CT data, with 34,561 samples for training, 6,491 samples for validation and 17,778 samples for testing. (5) PathMNIST (Kather et al., 2019) consisting of colon pathology data, with 89,996 samples for training, 10,004 samples for validation and 7,180 samples for testing.

BloodMNIST, OrganAMNIST, and PathMNIST datasets are derived from clinical data such as blood cell images, abdominal CT scans, and colon pathology slides, and are widely used in the medical AI literature. They naturally exhibit class imbalance since some disease categories occur much less frequently than others. This property makes them well suited for evaluating our Maverick-aware framework because they reflect realistic rare disease scenarios where Maverick clients who hold data from rare or underrepresented classes are especially important.

**Model.** We utilize a lightweight MLP neural network (Popescu et al., 2009) for MNIST and a commonly employed CNN (Albawi et al., 2017) for the CIFAR-10, BloodMNIST, OrganAMNIST, and PathMNIST dataset.

**Validation dataset.** For both MNIST and CIFAR-10 datasets, we follow common practice in existing Shapley-based FL methods (MR, TMR, and GTG) by using a balanced server-side validation set. For both MNIST and CIFAR-10, we randomly split 20% of testing samples as validation dataset. In the BloodM-

NIST, OrganAMNIST, and PathMNIST datasets, we utilize the provided validation set which follows the imbalanced distribution of the training samples.

**Maverick Scenarios.** Both MNIST and CIFAR-10 datasets are uniformly distributed across all 10 class labels. BloodMNIST, OrganAMNIST, and PathMNIST datasets are having imbalanced data distribution. These three datasets have different number of class labels, 8 class labels in BloodMNIST, 11 in OrganAM-NIST, and 9 in PathMNIST. In order to simulate heterogeneous settings, we adopt the widely used practical data heterogeneity setting (Zhang et al., 2023), where the heterogeneity is controlled by the Dirichlet distribution, denoted as $Dir(\gamma)$. Increasing $\gamma$ decreases the degree of heterogeneity. We vary $\gamma$ to control the level of heterogeneity and create three Maverick scenarios as follows: (i) Mavericks + $Dir(10)$: Most IID setting (the data distribution among non-Maverick clients is identical.) (ii) Mavericks + $Dir(1)$: Moderately heterogeneous setting, and (iii) Mavericks + $Dir(0.1)$: Most Non-IID setting, where data distribution among clients is highly diverse.

In all three settings, we have 50 clients with 10% selection rate of all clients in each FL round. For MNIST and CIFAR-10 datasets, we have 48 non-Mavericks and 2 Mavericks, and each Maverick exclusively owns one rare class as shown in Figure 1 (We randomly select class 5 and class 8 as Mavericks classes). For BloodMNIST, OrganAMNIST and PathMNIST datasets, we have 42 non-Mavericks and 8 Mavericks, each rare class is shared by 4 Mavericks as shown in Figures 7, 8, and 9. For these imbalanced datasets, we select the classes with the fewest samples as the Maverick classes, since these correspond to clinically important but underrepresented cases in practice. We then distribute these rare classes to a small number of Maverick clients, mimicking hospitals or labs that hold limited but critical data. For all settings, Mavericks not only have data samples from rare classes but also from the other classes. Figures 1, 7, 8, and 9 also illustrate the three heterogeneity scenarios, with $\gamma$ set to 10, 1, and 0.1, respectively. Additionally, we consider a 5 clients setting (4 non-Mavericks and 1 Maverick) without client selection (*i.e.*, all 5 clients participate in each round). In 5 clients setting, the data distribution of non-Mavericks is IID and the Maverick exclusively owns one class.

**Adversarial and Free-riding Scenarios.** In this setting, we have 58 clients (48 non-Mavericks, 2 Mavericks, 6 adversarial clients and 2 free-riders). We consider the existence of adversarial clients during the federated training process. They can mislead the model by injecting poisonous data or model updates. In our experiments, we include three types of malicious behaviors. (i) Label-flipping clients who flip the label of their dataset (Shen et al., 2023). (ii) Data-poisoned clients who participanting training with random generated data (Sandeepa et al., 2024). (iii) Update-poisoned clients who send random generated model updates to the server during the FL training (Xie et al., 2024). We also examine the existence of free riders (Chen et al., 2024). Free-riders do not update the model, but instead send the same model back to the server. Free-riders can enjoy the trained global model without any effort, whereas the benefits of benign users are greatly compromised.

**Baselines.** We consider seven client selection baselines: FedAvg (McMahan et al., 2017), S-FedAvg (Nagala-patti & Narayanam, 2021), FedEMD (Huang et al., 2022), FedProx (Li et al., 2020a), GreedyFed (Singhal et al., 2024), PoC (Cho et al., 2022), and HiCS-FL (Chen & Vikalo, 2024). FedAvg applies random sampling in each round. S-FedAvg and GreedyFed combine SV-based methods with client selection. S-FedAvg adopts MR (Song et al., 2019) and GreedyFed uses GTG (Liu et al., 2022) in their client selection. FedProx, PoC, and HiCS-FL propose mechanisms regarding data heterogeneity in FL. FedEMD combines EMD distance with client selection in the presence of Mavericks.

**FL Setup.** We train 100 rounds for MNIST and 200 for CIFAR-10, both with a batch size of 64. We employ 5 local training rounds for MNIST and a single local training round for CIFAR-10. We train BloodMNIST for 200 rounds and OrganAMNIST for 100 rounds. PathMNIST is trained for 100 rounds under the Mavericks + $Dir(10)$ and Mavericks + $Dir(1)$ settings, and 200 rounds under the Mavericks + $Dir(0.1)$ setting. Under the BloodMNIST, OrganAMNIST, and PathMNIST datasets, we have one local training round and use a batch size of 64. The learning rate is 0.05 for all baselines in all datasets. Our algorithm is implemented in Pytorch and we perform experiments on two NVIDIA RTX A5000 and two Xeon Silver 4316.

**Evaluation Metrics.** To assess the effectiveness of evaluated mechanisms, we consider the test accuracy, Macro-F1 score (averaged across all classes), and recall as the utility metrics. For the datasets with balanced

Table 1: Test accuracy (%) of different algorithms across three Mavericks scenarios on MNIST and CIFAR-10 dataset, with 50 clients under client selection.

| Settings | Mavericks + $Dir(10)$ | | Mavericks + $Dir(1)$ | | Mavericks + $Dir(0.1)$ | |
|---|---|---|---|---|---|---|
| Methods | MNIST | Cifar10 | MNIST | Cifar10 | MNIST | Cifar10 |
| FedAvg | 79.13±0.29 | 63.36±0.29 | 79.00±0.14 | 61.52±0.91 | 74.55±2.48 | 48.68±0.83 |
| FedProx | 78.53±1.17 | 59.84±0.93 | 77.54±0.30 | 57.07±2.04 | 75.08±0.32 | 41.06±6.58 |
| PoC | 91.27±0.71 | **67.07±0.98** | 78.76±0.23 | 60.18±4.26 | 54.63±2.62 | 54.77±6.38 |
| FedEMD | 85.51±0.23 | 55.84±0.40 | 78.05±0.96 | 55.66±0.58 | 65.39±2.93 | 47.18±0.90 |
| S-FedAvg | 80.66±2.30 | 62.98±0.29 | 82.56±1.38 | 61.42±0.44 | 77.24±0.82 | 45.94±3.94 |
| GreedyFed | 80.97±3.55 | 60.60±0.44 | 82.47±3.16 | 58.10±0.79 | **89.73±1.40** | 51.97±3.92 |
| **FedMS** | **92.96±0.32** | 65.92±0.68 | **92.67±0.10** | **65.92±0.75** | 89.62±0.57 | **59.75±2.10** |

Table 2: Macro-F1 scores of different algorithms under the Mavericks + $Dir(10)$ setting across three datasets (BloodMNIST, OrganAMNIST, and PathMNIST) with 50 clients under client selection. Values in parentheses indicate recall on rare classes.

| Methods | OrganAMNIST | PathMNIST | BloodMNIST |
|---|---|---|---|
| FedAVG | $0.65 \pm 0.01$ ($0.18 \pm 0.02$) | $0.51 \pm 0.03$ ($0.01 \pm 0.02$) | $0.47 \pm 0.12$ ($0.21 \pm 0.26$) |
| FedProx | $0.57 \pm 0.01$ ($0.07 \pm 0.05$) | $0.48 \pm 0.04$ ($0.00 \pm 0.00$) | $0.48 \pm 0.08$ ($0.21 \pm 0.23$) |
| POC | $0.66 \pm 0.02$ ($0.19 \pm 0.09$) | $0.51 \pm 0.02$ ($0.01 \pm 0.01$) | $0.50 \pm 0.04$ ($0.13 \pm 0.22$) |
| FedEMD | $0.67 \pm 0.01$ ($0.25 \pm 0.09$) | $0.55 \pm 0.02$ ($0.07 \pm 0.06$) | $\mathbf{0.51 \pm 0.03}$ ($0.00 \pm 0.00$) |
| GreedyFed | $0.65 \pm 0.03$ ($0.20 \pm 0.22$) | $0.50 \pm 0.02$ ($0.00 \pm 0.00$) | $0.50 \pm 0.01$ ($0.00 \pm 0.00$) |
| S-FedAVG | $\mathbf{0.68 \pm 0.01}$ ($0.29 \pm 0.06$) | $0.50 \pm 0.01$ ($0.02 \pm 0.04$) | $0.51 \pm 0.09$ ($0.04 \pm 0.07$) |
| HiCS-FL | $0.66 \pm 0.02$ ($0.38 \pm 0.05$) | $0.59 \pm 0.01$ ($0.25 \pm 0.11$) | $0.43 \pm 0.06$ ($0.17 \pm 0.29$) |
| **FedMS** | $0.66 \pm 0.04$ $\mathbf{(0.69\pm0.03)}$ | $\mathbf{0.66 \pm 0.06}$ $\mathbf{(0.68\pm0.26)}$ | $0.50 \pm 0.01$ $\mathbf{(0.35\pm0.03)}$ |

distribution like MNIST and CIFAR-10, we use test accuracy as the main metric, as shown in Table 1. For the datasets with imbalanced distribution including BloodMNIST, OrganAMNIST, and PathMNIST, we report recall of the Maverick classes and Macro-F1 score as our main evaluation metrics, as shown in Tables 2, 3, and 4. In addition, we evaluate different schemes based on their Shapley Rewards (SR) to the Mavericks. A larger SR is associated with higher contributions. We also utilize the participation rate (%) of Mavericks, adversaries, and free-riders as the robustness metric, as shown in Table 5.

**Hyperparameter Settings.** We tune the hyperparameters of all baselines using grid search. In FedProx, we set the weight of proximal term $\mu$ as 1 (selecting from [0.1, 1]). For PoC, we adopt the adaptive variant $\pi_{adapow-d}$ and set the decay factor as 0.9. In FedEMD, we set the distance coefficient (denoted by $\beta$ in that scheme Huang et al. (2022)) as 0.01. We choose $\alpha = 0.7$ for S-FedAvg and GreedyFed. In S-FedAvg, we utilize MR to calculate SV of all permutations (without using the Monte-Carlo sampling). For HiCS-FL, the scaling parameter T (temperature) used in data heterogeneity estimation was set to 0.0025. For our proposed FedMS, we choose temperature $T = 0.01$, $\alpha = 0.8$, and error threshold $\epsilon_b = 0.01$, $\epsilon_i = 0.001$. In each round, we calculate the class-wise accuracy difference and sum it to obtain $\hat{d}^t$, which measures the overall change in class-wise performance between the coreset $\hat{\mathcal{K}}^t$ and the initial global model of round $t$. If the model performance of coreset $\hat{\mathcal{K}}^t$ diverges significantly from the initial global model of round $t$ (i.e., $\hat{d}^t < 0$ and $|\hat{d}^t| > \epsilon_d$), the server discards the coreset and does not update the model for round $t$. We initially set $\epsilon_d$ to 3.0, and it exponentially decays to 0.1 by the end of training in all settings.

## 5.2 Evaluation Results

**Main Result I: FedMS Improves Model Performance.** Figure 2 illustrates how the test accuracy and SR change during training in the 5 client setting (without client selection). We evaluate the test accuracy under two settings: *Accuracy With All Clients* and *Accuracy Without Mavericks*. In Figure 2, there is a gap between the *Accuracy With All Clients* setting and the *Accuracy Without Mavericks* setting, and this gap reveals the importance of Mavericks. That is, without leveraging the Mavericks, the full potential of the model cannot be achieved, motivating our FedMS client selection mechanism.

Table 3: Macro-F1 scores of different algorithms under the Mavericks + $Dir(1)$ setting across three datasets (BloodMNIST, OrganAMNIST, and PathMNIST) with 50 clients under client selection. Values in parentheses indicate recall on rare classes.

| Methods | OrganAMNIST | PathMNIST | BloodMNIST |
|---|---|---|---|
| FedAVG | $0.62 \pm 0.05$ $(0.17 \pm 0.06)$ | $0.47 \pm 0.06$ $(0.00 \pm 0.00)$ | $\mathbf{0.55} \pm \mathbf{0.03}$ $(0.21 \pm 0.22)$ |
| FedProx | $0.54 \pm 0.05$ $(0.09 \pm 0.08)$ | $0.38 \pm 0.13$ $(0.00 \pm 0.00)$ | $0.51 \pm 0.03$ $(0.05 \pm 0.07)$ |
| POC | $0.63 \pm 0.03$ $(0.15 \pm 0.11)$ | $0.52 \pm 0.01$ $(0.02 \pm 0.02)$ | $0.42 \pm 0.10$ $(0.12 \pm 0.17)$ |
| FedEMD | $0.66 \pm 0.01$ $(0.30 \pm 0.10)$ | $0.51 \pm 0.04$ $(0.04 \pm 0.06)$ | $0.54 \pm 0.04$ $(0.12 \pm 0.14)$ |
| GreedyFed | $0.61 \pm 0.01$ $(0.00 \pm 0.00)$ | $0.51 \pm 0.03$ $(0.00 \pm 0.00)$ | $0.52 \pm 0.00$ $(0.00 \pm 0.00)$ |
| S-FedAVG | $0.64 \pm 0.04$ $(0.19 \pm 0.07)$ | $0.50 \pm 0.03$ $(0.06 \pm 0.09)$ | $0.50 \pm 0.06$ $(0.34 \pm 0.35)$ |
| HiCS-FL | $0.66 \pm 0.03$ $(0.38 \pm 0.12)$ | $0.57 \pm 0.04$ $(0.30 \pm 0.20)$ | $0.46 \pm 0.03$ $(0.00 \pm 0.00)$ |
| **FedMS** | $\mathbf{0.67} \pm \mathbf{0.02}$ $(\mathbf{0.54 \pm 0.07})$ | $\mathbf{0.64} \pm \mathbf{0.03}$ $(\mathbf{0.72 \pm 0.14})$ | $0.49 \pm 0.02$ $(\mathbf{0.41 \pm 0.41})$ |

Table 4: Macro-F1 scores of different algorithms under the Mavericks + $Dir(0.1)$ setting across three datasets (BloodMNIST, OrganAMNIST, and PathMNIST) with 50 clients under client selection. Values in parentheses indicate recall on rare classes.

| Methods | OrganAMNIST | PathMNIST | BloodMNIST |
|---|---|---|---|
| FedAVG | $0.50 \pm 0.06$ $(0.27 \pm 0.02)$ | $0.38 \pm 0.09$ $(0.07 \pm 0.08)$ | $0.36 \pm 0.13$ $(0.17 \pm 0.29)$ |
| FedProx | $0.44 \pm 0.05$ $(0.27 \pm 0.02)$ | $0.36 \pm 0.10$ $(0.09 \pm 0.15)$ | $0.31 \pm 0.05$ $(0.16 \pm 0.29)$ |
| POC | $0.42 \pm 0.14$ $(0.16 \pm 0.15)$ | $0.40 \pm 0.08$ $(0.04 \pm 0.05)$ | $0.32 \pm 0.03$ $(0.04 \pm 0.04)$ |
| FedEMD | $0.50 \pm 0.09$ $(0.22 \pm 0.20)$ | $0.48 \pm 0.05$ $(0.13 \pm 0.13)$ | $0.40 \pm 0.04$ $(0.09 \pm 0.16)$ |
| GreedyFed | $0.55 \pm 0.03$ $(0.33 \pm 0.12)$ | $0.41 \pm 0.05$ $(0.00 \pm 0.00)$ | $0.37 \pm 0.09$ $(0.00 \pm 0.00)$ |
| S-FedAVG | $0.52 \pm 0.00$ $(0.27 \pm 0.04)$ | $0.50 \pm 0.04$ $(0.14 \pm 0.07)$ | $\mathbf{0.48} \pm \mathbf{0.03}$ $(0.22 \pm 0.22)$ |
| HiCS-FL | $0.45 \pm 0.07$ $(0.10 \pm 0.08)$ | $0.43 \pm 0.01$ $(0.07 \pm 0.07)$ | $0.31 \pm 0.09$ $(0.21 \pm 0.36)$ |
| **FedMS** | $\mathbf{0.62} \pm \mathbf{0.02}$ $(\mathbf{0.39} \pm \mathbf{0.25})$ | $\mathbf{0.60} \pm \mathbf{0.12}$ $(\mathbf{0.48 \pm 0.42})$ | $0.48 \pm 0.11$ $(\mathbf{0.55 \pm 0.31})$ |

Comparing the accuracy of the proposed FedMS vs. the baselines for the MNIST and CIFAR-10 datasets under all three Maverick scenarios are given in Table 1. In all three scenarios from IID to Non-IID settings, our proposed FedMS maintains high accuracy and achieves better model performance than baselines in most settings. Among the baseline methods, only S-FedAvg and GreedyFed adopt SV in their client selection process but none of them considers the Mavericks settings. In those SV-based methods, the low SR of Mavericks decreases their selection probability during FL training, resulting in under-utilization of the Mavericks and poor performance of global model. Since FedMS effectively identifies the most contributing clients in a class-wise manner, it successfully selects the Mavericks and improves model performance. As shown in Tables 2, 3, and 4, results on the BloodMNIST, OrganAMNIST and PathMNIST datasets further demonstrate that our FedMS consistently improves Maverick-class accuracy, ensuring that rare or underrepresented classes are better learned and not overlooked, while maintaining strong overall model performance.

FedEMD applies a decreasing selection probability for the Mavericks as training rounds progress. As indicated in Figure 2, the accuracy gap between the *Accuracy With All Clients* setting and the *Accuracy Without Mavericks* setting increases as the training progresses to later rounds, demonstrating the need for selecting the Mavericks in later training rounds as well. Our approach differs from FedEMD by not relying on the distance of local and global data distributions in making client selection decisions. Instead, we prioritize the class-wise contribution of each client during the selection process, thus, our method achieves robust SR and improved accuracy compared to FedEMD. Additionally, querying local data distributions in FedEMD may lead to privacy leakage for clients, whereas our method does not require extra information from clients.

**Main Result II: FedMS Aligns Shapley Rewards (SR) with Actual Client Contributions.** FedMS computes SR that better reflect client contributions by leveraging class-wise SV and class difficulties $\boldsymbol{\beta}$. If rare classes owned by the Mavericks perform poorly on the validation dataset, our mechanism increases the class difficulty associated with these rare classes. Hence, our system provides SR that accurately capture the contributions of Mavericks.

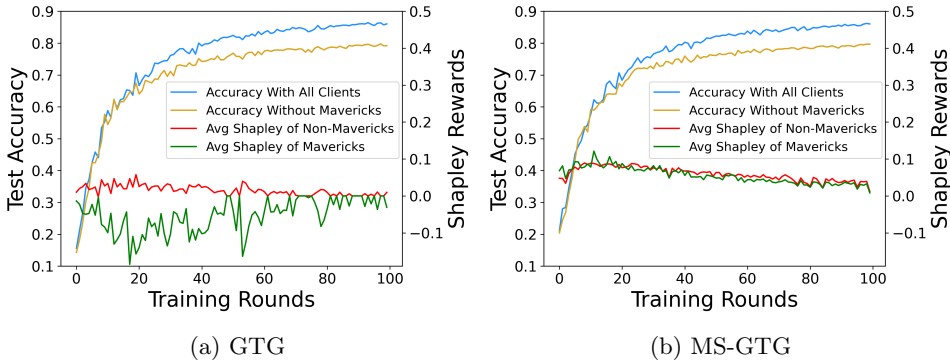

(a) GTG

(b) MS-GTG

Figure 2: Comparison of test accuracy and Shapley rewards with 5 clients (without client selection) for the MNIST dataset using GTG and MS-GTG under FedAvg.

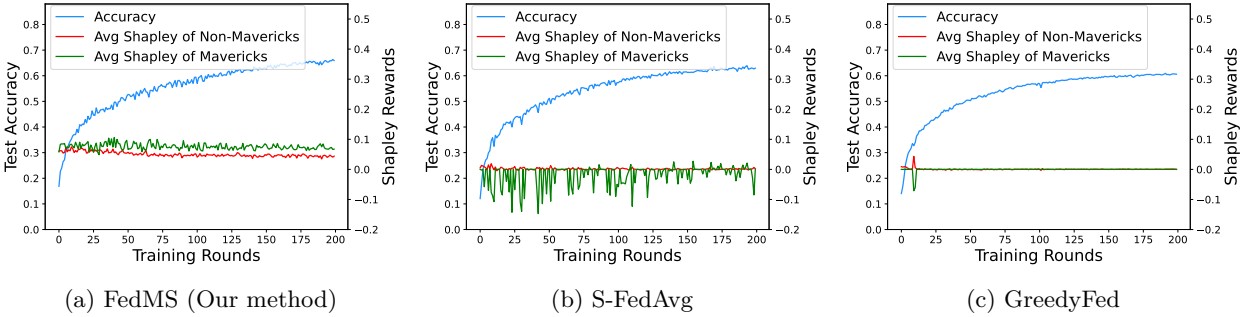

(a) FedMS (Our method)

(b) S-FedAvg

(c) GreedyFed

Figure 3: Comparison of test accuracy and Shapley rewards with 50 clients (with client selection) for the CIFAR-10 dataset under the Mavericks $+ Dir(10)$ (most IID) setting using MS-GTG for FedMS, MR for S-FedAvg and GTG for GreedyFed.

In Figure 2, we see that global model accuracy increases with the participation of the Mavericks. Despite their critical importance for improved accuracy, in Figure 4, when considering the scenario with 50 clients (with client selection), we observe that the average SR of the Mavericks is considerably lower than those of the non-Mavericks in S-FedAvg and GreedyFed when the SR are obtained directly from MR and GTG, respectively.[3] In contrast, Figure 4a exhibits a robust SR for Mavericks, reflecting their importance in later rounds for improved accuracy. In Figure 4a, the elevated SR for Mavericks show the superiority of MS-GTG over the standard GTG when Mavericks are involved in training. Even when there is no client selection, the main benefit of Maverick-Shapley is assigning SR that better reflect the contributions of Mavericks, as demonstrated in Figure 2(b).

When examining the SR, we observe from Figure 3 that FedMS allocates more rewards to Mavericks compared to non-Mavericks in the IID setting, in line with the observed accuracy benefit of training with the Mavericks. In contrast, S-FedAvg and GreedyFed provide lower rewards to Mavericks compared to non-Mavericks, as Mavericks are easily identified as outliers by Maverick-unaware Shapley methods like MR and GTG in the Mavericks $+ Dir(10)$ (most IID) setting.

In the Mavericks $+ Dir(0.1)$ (most Non-IID) setting, Figure 1c presents that the clients' data distributions are heterogeneous, with some non-Mavericks also possessing a large number of samples from specific classes. As shown in Figure 5, S-FedAvg and GreedyFed allocate lower rewards to non-Mavericks compared to Mavericks, even though the classes held by non-Mavericks are also important. In contrast, FedMS ensures a more contribution-aligned distribution of SR among Mavericks and non-Mavericks, as illustrated in Figure 5a. Our proposed FedMS assigns equal importance to all classes and distribute the SR proportionally based on each client's class contribution. Regarding the utility metric, we notice that FedMS better utilizes

---

[3]In our evaluation, we stick to original implementations of the baselines and use GTG (Liu et al., 2022) to calculate SR for GreedyFed, and MR (Song et al., 2019) for S-FedAvg.

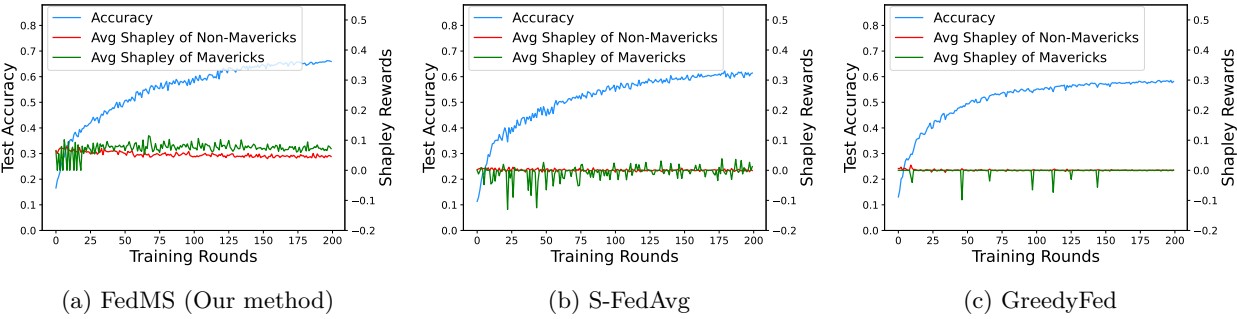

(a) FedMS (Our method)          (b) S-FedAvg          (c) GreedyFed

Figure 4: Comparison of test accuracy and Shapley rewards with 50 clients (with client selection) for the CIFAR-10 dataset under the Mavericks+$Dir(1)$ setting using MS-GTG for FedMS, MR for S-FedAvg and GTG for GreedyFed.

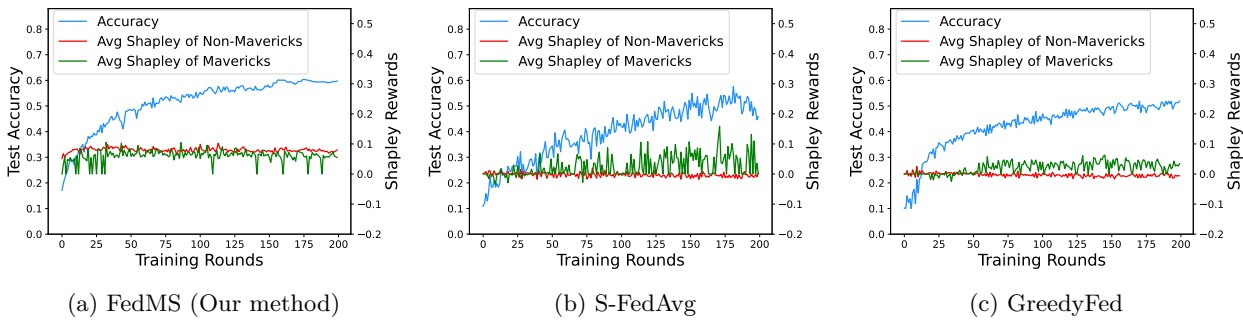

(a) FedMS (Our method)          (b) S-FedAvg          (c) GreedyFed

Figure 5: Comparison of test accuracy and Shapley rewards with 50 clients (with client selection) for the CIFAR-10 dataset under the Mavericks + $Dir(0.1)$ (most Non-IID) setting using MS-GTG for FedMS, MR for S-FedAvg and GTG for GreedyFed.

Mavericks, resulting in an overall improvement in model accuracy compared to S-FedAvg and GreedyFed on both Mavericks + $Dir(10)$ setting and Mavericks + $Dir(0.1)$ settings, as also demonstrated in Table 1.

**Main Result III: FedMS Shows Robustness to Adversaries and Free-riders.** Table 5 shows the participation rate (%) of Mavericks, adversaries and free-riders in FL baseline methods. A low participation rate of adversaries and free-riders is critical, since adversaries can inject harmful updates and free-riders contribute no useful updates, both of which undermine global training. Conversely, a high participation rate of Mavericks is desirable, as their data captures rare or underrepresented classes and directly improves generalization on these classes. As shown in Table 5, FedMS achieves this dual objective: it effectively suppresses adversaries and free-riders while maintaining strong inclusion of Mavericks. In contrast, baseline methods show a trade-off: some reduce adversarial and free-riders participation but fail to select Mavericks, while others increase Mavericks participation but allow adversaries and free-riders into training. This demonstrates that FedMS balances robustness to adversaries and free-riders with the inclusion of Mavericks.

## 6 Discussion

**Rewards as Incentives for Participation.** Throughout this paper, we utilized Mavericks in FL, so as to improve both the model performance and the rewards given to clients. Model performance is important from the server's perspective, which can be, *e.g.,* companies, hospitals, or institutions. Properly assessing the contribution and rewarding participants to participate in the FL training process, is equally important. Clients, like the Mavericks who own rare data, are valuable to the server and should receive rewards that reflect their contributions. Without proper incentives or rewards, Mavericks may refuse to join the FL training if their contributions are severely undervalued or, even worse, if they are deemed as outliers.

Table 5: The participation rate (%) of Mavericks, adversaries and free-riders in 7 FL methods under the MNIST dataset with Mavericks $+ Dir(0.1)$ setting.

| Settings | Label-flipping | Data-poisoned | Update-poisoned | Free-riding | Mavericks |
|---|---|---|---|---|---|
| FedAvg | 9.5% | 6.0% | 7.0% | 12.0% | 9.0% |
| FedProx | 9.0% | 10.5% | 8.5% | 9.5% | 6.5% |
| PoC | 8.0% | 9.5% | 8.0% | 9.0% | 8.0% |
| FedEMD | **0.0%** | 1.0% | **0.0%** | **0.5%** | 1.5% |
| S-FedAvg | 7.0% | 10.0% | 8.5% | 8.5% | 10.0% |
| GreedyFed | 1.0% | 60.0% | 1.0% | 42.5% | 47.0% |
| FedMS | 0.5% | **0.0%** | 1.0% | 2.5% | **54.0%** |

In this work, we aim to assign Shapley rewards and use class difficulty $\boldsymbol{\beta}$ to represent the importance of each class in each training round. Our approach encourages clients with valuable data to participate in the FL training. By recognizing the contributions of the Mavericks, we establish a contribution-guided rewarding mechanism for all participants in the FL system, as each potential client knows that their unique contributions would be properly acknowledged.

# 7 Conclusion

Client selection plays a pivotal role in FL, especially in the presence of Mavericks, as it allows for the optimization of the utility derived from diverse model updates. In this paper, we propose FedMS, a Maverick-aware Shapley valuation mechanism for client selection in FL that not only provides accurate evaluation of Mavericks contributions but also effectively selects the most contributing clients in each FL round. We show that our FedMS achieves better model performance and delivers a contribution-aligned distribution of Shapley rewards compared to state-of-the-art approaches, while providing robustness against adversaries and free-riders.

**Acknowledgments**

This work was partially supported by the National Science Foundation (NSF) under Awards 1956393 and 1900654 at UC Irvine, and Award 1956435 at USC.

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

## A Appendix

This appendix extends the main body of our paper, providing supplementary materials to enhance the understanding of our proposed method. This appendix first introduces additional Maverick-aware Shapley approximations, namely MS-MR and MS-TMR. Then, we provide data samples distribution for BloodMNIST, OrganAMNIST, and PathMNIST datasets applied in the paper. Table 6 presents the main parameters and notations of our proposed method and Figure 6 illustrates the FL framework in the presence of Mavericks.

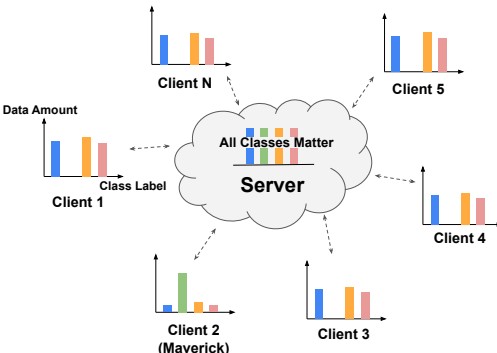

Figure 6: FL framework in the presence of Mavericks. In this example, client 2 is a Maverick, as only that client possesses samples from the green label.

### A.1 Maverick-Aware SV Approximations

---

**Algorithm 3:** Maverick-Shapley MR (MS-MR)

---

**1** **Input**: Updated client models $\{\boldsymbol{w}_i^t\}_{i\in\mathcal{K}^t}$; current server model $\boldsymbol{w}^t$; validation dataset at server $\mathcal{D}_{\text{val}}$; class-wise accuracy function $\mathcal{V}_{class}(\cdot)$; $\mathcal{M}$: set of class labels; $\mathcal{K}^t$: set of clients in round $t$;.

**2** **Hyperparameter:** Temperature T

**3** **Initialize:** $\phi_i = 0, \forall i \in \mathcal{K}^t$

**4** **for** *each subset $Q \subseteq \mathcal{K}^t$* **do**

**5**    $\widetilde{\boldsymbol{w}}_Q = \texttt{ModelAverage}(\{\boldsymbol{w}_i^t\}_{i\in Q}, \boldsymbol{w}^t)$

**6** **for** *client $i \in \mathcal{K}^t$* **do**

**7**    **for** *class $c \in \mathcal{M}$* **do**

**8**        $\phi_i^c = \frac{1}{|\mathcal{K}^t|} \sum\limits_{\mathcal{Q}\subseteq\mathcal{K}^t\backslash\{i\}} \frac{\mathcal{V}_{class}^c(\widetilde{\boldsymbol{w}}_{\mathcal{Q}\cup\{i\}};\mathcal{D}_{\text{val}}) - \mathcal{V}_{class}^c(\widetilde{\boldsymbol{w}}_{\mathcal{Q}};\mathcal{D}_{\text{val}})}{\binom{|\mathcal{K}^t|-1}{|\mathcal{Q}|}}$

**9** *// Find the coreset $\hat{\mathcal{K}}^t$ and its class-wise accuracy $\hat{v}$*

**10** $\hat{\mathcal{K}}^t, \hat{v} \leftarrow \text{argmax}_{Q\subseteq\mathcal{K}^t} \sum\limits_{c\in\mathcal{M}} \mathcal{V}_{class}^c(\widetilde{\boldsymbol{w}}_Q, \mathcal{D}_{\text{val}})$

**11** *// Obtain class difficulty $\boldsymbol{\beta}$*

**12** $\beta^c = \frac{exp(\frac{1-\hat{v}^c}{T})}{\sum\limits_{c\in M} exp(\frac{1-\hat{v}^c}{T})}, \forall c \in M$

**13** **return** $\boldsymbol{\phi}, \boldsymbol{\beta}, \hat{\mathcal{K}}^t$

---

**Maverick-Shapley MR (MS-MR).** MS-MR is built on top of the Multi-Round (MR) approach (Song et al., 2019). In MR, the SV is calculated in every FL round and the subsets of client models are reconstructed by using the corresponding gradient updates. In our MS-MR, the server first calculates the class-wise SV (Line 8 in Algorithm 3) for each client, and then computes the contribution score $\hat{S}_i$ of each client $i$ with

---

**Algorithm 4:** Maverick-Shapley TMR (MS-TMR)

---

1   **Input**: Updated client models $\{\boldsymbol{w}_i^t\}_{i \in \mathcal{K}^t}$; current server model $\boldsymbol{w}^t$; validation dataset at server $\mathcal{D}_{\text{val}}$; class-wise accuracy function $\mathcal{V}_{class}(\cdot)$; $\mathcal{M}$: set of class labels; $\mathcal{K}^t$: set of clients in round $t$;.

2   **Hyperparameter:** Error threshold $\epsilon_b$, Temperature T

3   **Initialize:** $\phi_i = 0, \forall i \in \mathcal{K}^t$

4   $v_0 = \mathcal{V}_{class}(\boldsymbol{w}^t; \mathcal{D}_{\text{val}})$, $v_N = \mathcal{V}_{class}(\boldsymbol{w}^{t+1}; \mathcal{D}_{\text{val}})$,

5   *// between round truncation*

6   **if** $|v_N - v_0| > \epsilon_b$ **then**

7     **for** *each subset $Q \subseteq \mathcal{K}^t$* **do**

8       $\widetilde{\boldsymbol{w}}_Q = \texttt{ModelAverage}(\{\boldsymbol{w}_i^t\}_{i \in Q}, \boldsymbol{w}^t)$

9     **for** *client $i \in \mathcal{K}^t$* **do**

10       **for** *class $c \in \mathcal{M}$* **do**

11         $\phi_i^c = \frac{1}{|\mathcal{K}^t|} \sum\limits_{\mathcal{Q} \subseteq \mathcal{K}^t \setminus \{i\}} \frac{\mathcal{V}_{class}^c(\widetilde{\boldsymbol{w}}_{\mathcal{Q} \cup \{i\}}; \mathcal{D}_{\text{val}}) - \mathcal{V}_{class}^c(\widetilde{\boldsymbol{w}}_{\mathcal{Q}}; \mathcal{D}_{\text{val}})}{\binom{|\mathcal{K}^t| - 1}{|\mathcal{Q}|}}$

12   *// Find the coreset $\hat{\mathcal{K}}^t$ and its class-wise accuracy $\hat{v}$*

13   $\hat{\mathcal{K}}^t, \hat{v} \leftarrow \text{argmax}_{Q \subseteq \mathcal{K}^t} \sum\limits_{c \in \mathcal{M}} \mathcal{V}_{class}^c(\widetilde{\boldsymbol{w}}_Q, \mathcal{D}_{\text{val}})$

14   *// Obtain class difficulty $\boldsymbol{\beta}$*

15   $\beta^c = \frac{exp(\frac{1 - \hat{v}^c}{T})}{\sum\limits_{c \in M} exp(\frac{1 - \hat{v}^c}{T})}, \forall c \in M$

16   **return** $\boldsymbol{\phi}, \boldsymbol{\beta}, \hat{\mathcal{K}}^t$

---

updated *class difficulty* $\boldsymbol{\beta}$ (Line 12 in in Algorithm 3) and the class-wise SVs $\phi^c$. In each round, the server selects the subset of clients $\hat{\mathcal{K}}^t$ that achieve the highest accuracy increase and aggregates only their updates as the updated global model of the next round.

**Maverick-Shapley TMR (MS-TMR).** MS-TMR is built on top of the Truncated Multi-Rounds Construction (TMR) approach (Wei et al., 2020). The TMR extends the MR by introducing a decay parameter $\lambda$ to eliminate unnecessary subset model reconstructions. In our MS-TMR, we introduce an error threshold $\epsilon_b$ to eliminate the unnecessary reconstructions (Line 2 in Algorithm 4).

Table 6: Notation and key parameters used in the paper.

| Notation | Description |
|---|---|
| $(\boldsymbol{x}, y)$ | Data sample with feature vector $\boldsymbol{x}$ and label $y$ |
| $\mathcal{K}$ | Set of clients, $\mathcal{K} = \{1, 2, \dots, I\}$ |
| $\mathcal{M}$ | Set of class labels, $\mathcal{M} = \{1, 2, \dots, C\}$ |
| $\eta_i$ | Learning rate of client $i$ |
| $\mathcal{D}_i$ | Local dataset of client $i$ |
| $\mathcal{D}_{val}$ | Validation dataset at the server |
| $\boldsymbol{w}$ | Global model weights |
| $\boldsymbol{w}_i$ | Local model weights of client $i$ |
| $\mathcal{V}_{class}(\cdot)$ | Class-wise accuracy function |
| $\boldsymbol{\phi}$ | Class-wise Shapley value vector for all clients |
| $\phi_i$ | Class-wise Shapley value of client $i$ |
| $\boldsymbol{\beta}$ | Class difficulty vector across all classes |
| $\beta^c$ | Class difficulty for class $c$ |
| $S_i^c$ | Accumulated Shapley value of client $i$ for class $c$ |
| $\alpha$ | Decay factor for accumulated Shapley value $S_i^c$ |
| $\hat{S}_i^c$ | Contribution score of client $i$ for class $c$ |
| $P_{\mathcal{S}}$ | Client selection probability vector over all clients |
| $P_{\mathcal{S},i}$ | Client selection probability of client $i$ |
| $T$ | Number of FL training rounds |
| $t$ | Index of FL round, $t = 0, 1, \dots, T-1$ |
| $E$ | Number of local epochs |
| $B$ | Mini-batch size |
| $\boldsymbol{w}^t$ | Global model weights in round $t$ |
| $\boldsymbol{w}_i^t$ | Local model weights of client $i$ in round $t$ |
| $\mathcal{K}^t$ | Set of selected clients in round $t$ with selection strategy $\pi$ |
| $\hat{\mathcal{K}}^t$ | coreset with the highest class-accuracy in round $t$ |
| $n_i^t$ | Dataset size of client $i$ in round $t$ |

## A.2 Data Samples Distribution for BloodMNIST, OrganAMNIST and PathMNIST Datasets

Figure 7 shows the data distribution in OrganAMNIST dataset across three Mavericks scenarios, with class 1 and class 2 as Mavericks classes. Figure 8 presents the data distribution in PathMNIST dataset across three Mavericks scenarios, with class 4 and class 6 as Mavericks classes. Figure 9 shows the data distribution in BloodMNIST dataset across three Mavericks scenarios, with class 0 and class 4 as Mavericks classes.

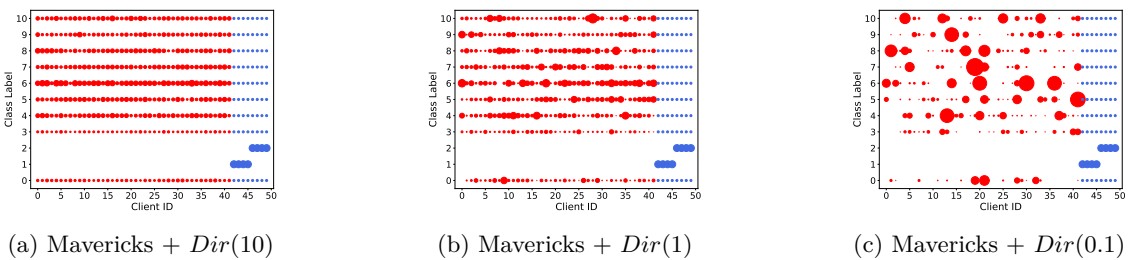

(a) Mavericks + $Dir(10)$      (b) Mavericks + $Dir(1)$      (c) Mavericks + $Dir(0.1)$

Figure 7: Data sample distribution across clients in three Mavericks scenarios for OrganAMNIST dataset. Mavericks are shown as blue circles, and non-Mavericks as red. Circle size indicates sample count, with heterogeneity controlled by $\gamma$ in $Dir(\gamma)$. Higher $\gamma$ means lower heterogeneity. (a) is the most IID setting with identical distributions for all but Mavericks. (b) shows moderate heterogeneity, and (c) is the most Non-IID. In all cases, the Mavericks exclusively hold certain class labels (e.g., class 1 and class 2), while still containing samples from other classes.

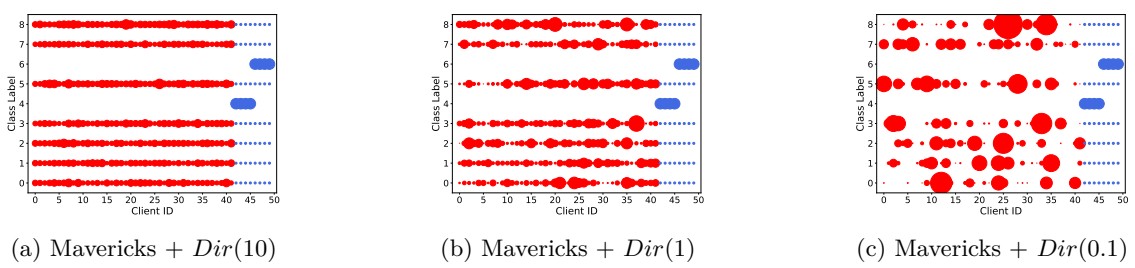

(a) Mavericks + $Dir(10)$      (b) Mavericks + $Dir(1)$      (c) Mavericks + $Dir(0.1)$

Figure 8: Data sample distribution across clients in three Mavericks scenarios for PathMNIST dataset. Mavericks are shown as blue circles, and non-Mavericks as red. Circle size indicates sample count, with heterogeneity controlled by $\gamma$ in $Dir(\gamma)$. Higher $\gamma$ means lower heterogeneity. (a) is the most IID setting with identical distributions for all but Mavericks. (b) shows moderate heterogeneity, and (c) is the most Non-IID. In all cases, the Mavericks exclusively hold certain class labels (e.g., class 4 and class 6), while still containing samples from other classes.

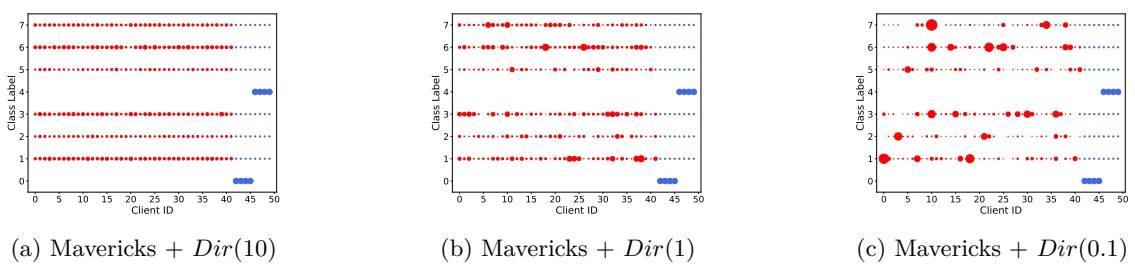

(a) Mavericks + $Dir(10)$      (b) Mavericks + $Dir(1)$      (c) Mavericks + $Dir(0.1)$

Figure 9: Data sample distribution across clients in three Mavericks scenarios for BloodMNIST dataset. Mavericks are shown as blue circles, and non-Mavericks as red. Circle size indicates sample count, with heterogeneity controlled by $\gamma$ in $Dir(\gamma)$. Higher $\gamma$ means lower heterogeneity. (a) is the most IID setting with identical distributions for all but Mavericks. (b) shows moderate heterogeneity, and (c) is the most Non-IID. In all cases, the Mavericks exclusively hold certain class labels (e.g., class 0 and class 4), while still containing samples from other classes.

