# OpenReview forum: "Rewarding the Rare: Maverick-Aware Shapley Valuation in Federated Learning"
_TMLR — Accepted by TMLR_

### Review · Reviewer_jgNt · 2025-09-04

**Summary Of Contributions:**

This paper investigates the problem of maverick-aware Shapley valuation (SV) in federated learning (FL). It argues that existing SV methods struggle to properly weight maverick clients, leading to suboptimal client selection and consequently weaker generalization performance. To address this limitation, the paper proposes using a class-wise recall ratio as the utility function in SV computations. Experiments conducted on the MNIST and CIFAR-10 datasets demonstrate the effectiveness of the proposed approach.

**Additional Comments:**

I have several additional concerns:
1. The first claimed contribution lacks sufficient supporting evidence.
2. The proposed use of a class-wise recall ratio to address imbalance re-weighting among categories across clients is simple and straightforward, but may not be sufficiently novel.
3. More recent federated learning approaches should be included for comparison.
4. Equation (7) could be directly referred to as the recall ratio for clarity.

**Audience:**

Yes

**Audience Explanation:**

It appears that maverick-aware Shapley valuation has not been extensively studied, yet it represents an important direction for FL.

**Broader Impact Concerns:**

N.A.

**Claims And Evidence:**

No

**Claims Explanation:**

Although the first claimed contribution is intuitive, it still lacks sufficient support. Additional empirical or theoretical evidence should be provided.

**Requested Changes:**

I have several additional concerns:
1. The first claimed contribution lacks sufficient supporting evidence.
2. The proposed use of a class-wise recall ratio to address imbalance re-weighting among categories across clients is simple and straightforward, but may not be sufficiently novel.
3. More recent federated learning approaches should be included for comparison.
4. Equation (7) could be directly referred to as the recall ratio for clarity.

---

> ### Author Response · Authors · 2025-10-03
> **Response to Reviewer jgNt**
>
> We thank the reviewer for the insightful comments and suggestions. Our responses to the reviewer’s comments are summarized below:
>
> **Regarding the first claimed contribution:** Our first contribution is to identify the limitation of uniform validation weighting in existing Shapley-based FL methods. This is not only intuitive but also supported empirically in our paper. Uniform weighting aggregates validation accuracy across classes as if each class were equally frequent and equally represented, which systematically disadvantages clients that hold rare or underrepresented classes. Such clients appear to contribute less, not because their data is less valuable, but because uniform weighting fails to capture their unique role in improving generalization on underrepresented classes.
>
> Our experiments directly demonstrate this: for example, in Figure 2, excluding Mavericks causes a clear accuracy drop, yet Mavericks consistently receive lower Shapley rewards under uniform weighting. Our new results (See Tables A, B and C in common comments) further confirm this limitation. Shapley-based baselines like GreedyFed (Using GTG) and S-FedAvg (Using MR) consistently yield low Maverick-class accuracy, even when their overall F1 score is high. For example, in PathMNIST and BloodMNIST under Dir(10) (**See Table A in common comments**), both methods fail to capture Mavericks’ contributions. In contrast, FedMS obviously improves Maverick-class performance. Together, these results provide empirical evidence that uniform weighting systematically misvalues Mavericks and limits model generalization.
>
> This observation aligns with [R4], which also reports that SV-based contribution measurements underestimate the role of Mavericks or clients with skewed data distributions.
>
> [R4] Jiyue Huang, Chi Hong, Yang Liu, Lydia Y Chen, and Stefanie Roos. Tackling Mavericks in Federated Learning via Adaptive Client Selection Strategy. In Proceedings of the AAAI Conference on Artificial Intelligence, 2022.
>
> **On novelty of the class-wise recall ratio**, we emphasize that while the formulation may appear simple, its integration with Shapley valuation is non-trivial and crucial. Existing SV methods assume uniform per-class weighting, which systematically misvalues Mavericks due to class imbalance. That is, during naive Shapley value computation, these Maverick clients appear to be contributing less than the majority of the participants as accuracy is considered as a whole. However, in our approach by computing the SV class-wise together with class difficulties, we are able to get a more granular valuation of the participating users (Mavericks and non-Mavericks alike). In essence, by directly incorporating class difficulty through recall-based weighting, our approach better captures each client’s true contribution.
>
> **Regarding comparisons with recent FL methods:** In addition to the existing six baselines (FedAvg, FedProx, PoC, FedEMD, S-FedAvg, GreedyFed), we have also included HiCS-FL [R1] as a recent client selection method that explicitly addresses data heterogeneity. This ensures that our evaluation covers both foundational methods and the latest state-of-the-art approaches.
>
> [R1]: Chen, Huancheng, and Haris Vikalo. "Heterogeneity-guided client sampling: Towards fast and efficient non-iid federated learning." Advances in Neural Information Processing Systems (NeurIPS 2024).
>
> We now provide evaluation results for additional diverse real-world medical imaging datasets and also include baselines. See responses to common comments above **CC1-DATASET and CC2-VALIDATION DATASET**. As shown in **Tables A, B, and C** in common comments, these new experimental results demonstrate that our FedMS consistently improves Maverick-class accuracy, ensuring that rare or underrepresented classes are better learned and not overlooked, while maintaining strong overall results and outperforming baseline methods.
>
> **On clarity of Equation (7)**, we agree and will explicitly refer to it as the class-wise recall ratio for improved clarity and readability.

---

### Review · Reviewer_ku3e · 2025-09-11

**Summary Of Contributions:**

This paper addresses a key challenge in Federated Learning (FL) and Shapley Value (SV): fairly evaluating the contributions of clients, particularly those holding rare or underrepresented data (referred to as Mavericks). Existing SV-based methods rely on uniformly weighted validation accuracy, which systematically undervalues Mavericks because rare classes are harder to learn and typically achieve lower accuracy. To address this issue, the paper proposes a Maverick-aware client selection algorithm that prioritizes clients based on their Maverick-Shapley contribution scores. The method adaptively selects clients and forms a coreset to maximize class-wise validation accuracy.

**Audience:**

Yes

**Audience Explanation:**

The investigated question about federated learning and data shapley value is of the interest of TMLR's audience

**Claims And Evidence:**

No

**Claims Explanation:**

The submission provides clear and well-documented empirical evidence that FedMS improves model accuracy, better aligns Shapley rewards with true contributions, and reduces adversarial and free-rider participation on MNIST and CIFAR-10 under synthetic Maverick scenarios. The algorithms are precisely specified, and results are reported with mean ± standard deviation across multiple baselines.


However, the evidence is only partially convincing due to its narrow scope.

1. The experiments are limited to small datasets such as MNIST and CIFAR-10, lack diversity in experimental settings, and do not include a scalability analysis.

2. The approach also assumes access to a central validation set containing all classes, which may be unrealistic in real-world FL deployments.

3. Moreover, the claim that the method is “private” is not well supported. Federated learning itself, as well as Shapley value evaluation, are not inherently privacy-preserving [A, B]. To substantiate this claim, one would need to demonstrate resistance to privacy attacks or integrate privacy-enhancing mechanisms such as differential privacy.

Overall, the claims are supported within the reported experimental setting, but the work would benefit from broader and more rigorous evaluation to be fully compelling.


[A] Boenisch et al., Reconstructing Individual Data Points in Federated Learning Hardened with Differential Privacy and Secure Aggregation, IEEE S&P 2023.


[B] Wang et al., Threshold KNN-Shapley: A Linear-Time and Privacy-Friendly Approach to Data Valuation, NeurIPS 2023.

**Requested Changes:**

1.Include larger and more diverse datasets, test with more clients and classes, and report runtime and communication overhead. Moreover, conducting experiments in a fully decentralized setting with data heterogeneity would further strengthen the results.

2. Discuss or propose alternatives to assuming a server-side validation set with full class coverage.


3. Clarify privacy claims: either remove them or justify them with privacy attack–resilience evaluations or by integrating privacy-preserving mechanisms (e.g., membership inference attacks, differential privacy, secure aggregation).

---

> ### Author Response · Authors · 2025-10-03
> **Response to Reviewer ku3e**
>
> We thank the Reviewer ku3e for the valuable comments. We provide clarifications and responses to the raised concerns as follows:
>
> **Regarding dataset realism and evaluation:** See response in common comments above “**CC1-DATASET**”. We now provide results for additional diverse real-world medical imaging datasets and baselines.
>
> **On validation set assumptions:**  See response in common comments above “**CC2-VALIDATION DATASET**”.  We include additional experiments with imbalanced validation sets, showing that our approach remains robust even under imbalanced or skewed validation splits.
>
> **On privacy**, we indeed do not make any privacy contributions in this work. Our method focuses on robustness in client valuation, which is orthogonal to Privacy-Enhancing Technologies (PETs) for FL. We will revise the manuscript to avoid overstating privacy and also refer to the suggested papers [R2, R3] as examples of privacy-enhancing mechanisms that can be built on top of our method.
>
> [R2] Boenisch et al., Reconstructing Individual Data Points in Federated Learning Hardened with Differential Privacy and Secure Aggregation, IEEE S&P 2023.
>
> [R3] Wang et al., Threshold KNN-Shapley: A Linear-Time and Privacy-Friendly Approach to Data Valuation, NeurIPS 2023.

---

### Review · Reviewer_T9fC · 2025-09-20

**Summary Of Contributions:**

The paper proposes Maverick-Shapley, a variant of Shapley value estimation for federated learning.
It quantifies each client’s contribution based on weighted per-class accuracy, where the weights are determined by class difficulty. The class difficulty is obtained by normalizing the class accuracies.
This approach is designed to address situations where “Maverick” clients, who hold rare or unique classes, are undervalued under standard Shapley-based mechanisms.
The method is further extended into a client selection strategy called FedMS and evaluated on MNIST and CIFAR-10.

## Strengths
- The motivation is clear (standard shapley undervalues rare-class clients in principle)
- Solution is simple and straightforward
- The paper is written very clearly
- I especially like the experiments with adversarial clients, which can illustrate the robustness of the proposed approach

## Weakness
### Strong assumptions
- Validation dataset assumptions: The approach relies on a balanced and representative validation set. If the validation dataset is biased or non-representative, the class-difficulty weights may be misleading as the class-difficulty weight depends on the accuracy.
- Nature of class imbalance: This paper essentially addresses class imbalance. In federated learning, there are two main types:
-- Within-client imbalance – each client has a skewed class distribution (e.g., ratio of positive to negative is 1:10).
-- Cross-client imbalance – one or a few clients hold all or most samples of a single class, while others hold the rest.
This paper focuses on case (2). In practice, this is a very strong assumption: for example, it is rare for a hospital to only have patients with a single rare disease, unless it is a specialized clinic. I would appreciate if the authors could justify the realism of this problem setting with evidence or references.
### Realism and Evaluation limitations:
- Scope of applicability: Maverick-Shapley is most helpful in case (2), where one client owns a unique class. In case (1), the benefit is less clear, since all classes are already present across clients.
- Alternative strategies: In the motivating scenarios (rare disease or rare accent), one could question whether federated learning is the right solution. Training specialized models for rare groups may sometimes be preferable, unless the explicit goal is to produce a single global model.
- Dataset realism: The paper demonstrates performance only on MNIST and CIFAR-10. Its effectiveness on real-world rare-class scenarios is therefore less justified.
### Results
- The results section feels monotonous and a bit repetitive— Figures 2, 3, 4, and 5 present similar trends. It is difficult to judge whether the model should allocate more weight to Mavericks or non-Mavericks, the presentation style makes the trends look very similar across methods and heterogeneity levels. If possible, the authors should provide confidence intervals to illustrate the statistical significance of the Shapley reward results between Mavericks and Non-Mavericks.
### Claim and evidence
Clarity of claims: Several statements in the paper are vague and would benefit from clarification. For example:
“When examining the SR, we observe from Figure 3 that FedMS allocates more rewards to Mavericks compared to non-Mavericks in the IID setting, in line with the observed accuracy benefit of training with the Mavericks.” → How exactly is this **in line** with the observed accuracy benefit? Please explain the causal link.
“Our proposed FedMS assigns equal importance to all classes and distributes the SR proportionally based on each client’s class contribution.” → This conclusion is not obvious from Figure 5a. Please justify how the figure supports this statement.
## Summary
While the motivation is clear and the empirical results are promising, the realism of the simulated scenarios, the reliance on strong validation set assumptions, the lack of clarification for some of the statements made in this paper, and the limited evaluation on real-world data raise concerns. Addressing these issues would significantly strengthen the paper and broaden its applicability

**Audience:**

Yes

**Audience Explanation:**

Interesting problem. However, I would like to see whether the exact scenario the authors simulate using MNIST and CIFAR10 to validate the proposed approach actually matches real-world scenarios.

**Claims And Evidence:**

No

**Claims Explanation:**

Please see the weakness section

**Requested Changes:**

See above

---

> ### Author Response · Authors · 2025-10-03
> **Response to Reviewer T9fC**
>
> Thanks Reviewer T9fC for the constructive feedback. We address the main concerns as follows.
>
> **Regarding dataset realism and evaluation:** See response in common comments above “**CC1-DATASET**”. We now provide results for additional diverse real-world medical imaging datasets and baselines.
>
> **On validation set assumptions:**  See response in common comments above “**CC2-VALIDATION DATASET**”.  We include additional experiments with imbalanced validation sets, showing that our approach remains robust even under imbalanced or skewed validation splits.
>
> **Clarification on the nature and realism of class imbalance:** To clarify our setup, Mavericks exclusively own certain rare classes but also contain other classes, as illustrated in Figure 1 of the manuscript. For example, in a medical setting, a Maverick (i.e., a clinic or hospital) may hold rare disease cases that no other participating client has such as Amyotrophic Lateral Sclerosis (ALS), but also still contain patients with common diseases. Thus, our assumption does not force any client to hold only a single class. As the Reviewer mentions, the main motivation is that certain clients inherently own data that is not represented elsewhere, and their lack of participation during training directly impacts the trained model’s performance on these classes.
>
> **Clarification on the scope of applicability:** We agree that Maverick-Shapley is most impactful under cross-client imbalance (case 2), which is what we focus on in this work. As mentioned, this case is critical in practice as unique and rare data not only helps with generalizability of the trained model but also contributes to having inclusive and fair models. Without incorporating Mavericks, it is not possible to achieve high accuracy on the rare classes they own.
>
> **Regarding the alternative strategies:** We want to highlight that Mavericks in our scenario possess rare but relevant data. That is, classes contained in these users help with the global model training. Without them, the global model cannot perform well on these rare classes, hampering its performance. Thus, the main goal is to employ a federated paradigm so that other users can make use of Maverick users’ data as well. As the Reviewer rightfully points out, specialized training for rare groups is a viable option if the goal is not to train a global model that can work well on all classes and serve the entire population.
>
> **On clarity of claims:**
>  In Figure 3, the increase in Shapley rewards (SR) for Mavericks coincides with improved test accuracy when they participate. The causal link is that Mavericks uniquely contribute rare classes, so allocating them higher SR directly reflects their essential role in raising overall accuracy (as without them the model cannot perform well on rare Maverick classes). Without their updates, the accuracy curve drops (as shown in Figure 2, “without Mavericks”), which confirms this alignment. We will revise the text to make this statement more clear.
>
>  **On clarity of claims:** Regarding Figure 5a, We agree that our earlier statement was unclear. Figure 5 corresponds to the extreme Non-IID setting, where the dataset distribution makes it quite challenging to perform federated learning. In this scenario, not only Mavericks but also some non-Mavericks contribute hard-to-learn classes such that prioritizing Mavericks alone is not enough for good accuracy performance. In this case, FedMS adaptively increases SR for all clients that contribute to difficult classes, rather than privileging only the Maverick-held classes. By contrast, Figures 5b and 5c show poor allocation of rewards, which leads to lower overall accuracy. FedMS achieves a more proportional alignment of SR with actual contributions, which explains the accuracy gains in Figure 5a compared to Figures 5b and 5c.

---

### Author Response · Authors · 2025-10-03
**Response to Common Comments of Reviewers**

Thank you to all reviewers for the thorough and valuable reviews. We provide responses to each reviewer individually.

In addition, here we provide answers and additional results for comments that were common across several reviewers (which we refer to as “**common comments**” or “**CC**”). In particular, we would like to highlight that we considered additional datasets and baselines (and report the results in **new Tables A, B, and C below**). If the paper is accepted, we can include these additional results in the final version.

**CC1-DATASET:**  Several reviewers asked about dataset realism and evaluation: In the initial submission, we use the standard datasets, MNIST and CIFAR-10, used in the Shapley value literature (e.g., MR (Song et al., 2019), TMR (Wei et al., 2020), and GTG (Liu et al., 2022)).  Based on your comment, we expanded our experiments to include three more diverse real-world medical imaging datasets, BloodMNIST, OrganAMNIST, and PathMNIST. These datasets are derived from clinical data such as blood cell images, abdominal CT scans, and colon pathology slides, and are widely used in the medical AI literature. They naturally exhibit class imbalance since some disease categories occur much less frequently than others. This property makes them well suited for evaluating our Maverick-aware framework because they reflect realistic rare disease scenarios where Maverick clients who hold data from rare or underrepresented classes are especially important. We adopt Dirichlet-based data partitions to control the degree of heterogeneity among clients: Mavericks +  Dir (10) corresponds to a highly IID setting, Mavericks + Dir (1) represents moderate non-IID heterogeneity, and Mavericks + Dir (0.1) denotes a highly non-IID setting.

To define Mavericks in these new datasets, we select the classes with the fewest samples as the Maverick classes, since these correspond to clinically important but underrepresented cases in practice. We then distribute these rare classes to a small number of Maverick clients, mimicking hospitals or labs that hold limited but critical data. Due to the imbalance in data distribution, we report Macro-F1 (averaged across all classes) and recall of the Maverick classes as our main evaluation metrics. In the new Tables A, B, and C, the first value corresponds to the Macro-F1 score (mean ± standard deviation), while the value in parentheses shows the recall of the Maverick classes (mean ± standard deviation). For example, on the OrganAMNIST dataset with the FedMS algorithm, an entry such as 0.67 ± 0.02 (0.54 ± 0.07) denotes a Macro-F1 score of 0.67 ± 0.02, with the value in parentheses representing the Maverick-class recall of 0.54 ± 0.07.

These new experimental results demonstrate that our FedMS consistently improves Maverick-class accuracy, ensuring that rare or underrepresented classes are better learned and not overlooked, while maintaining strong overall results and outperforming baseline methods.

**CC2-VALIDATION DATASET:**  In the initial submission, we follow common practice in existing Shapley-based FL methods (MR, TMR, and GTG) by using a balanced server-side validation set. However, our method does not force a balanced validation set with all classes. In the new experiments with additional datasets (see Tables A, B, and C), we construct the validation set according to the real, imbalanced distribution of the training samples. Unlike the earlier experiments, the server-side validation set is no longer balanced in these new experiments. By computing class-wise accuracy and adaptively updating the class difficulty weights (𝛽) in each round, our approach remains robust even under imbalanced or skewed validation splits. In fact, our design reduces dependence on a uniformly balanced validation set, since the relative per-class performance, rather than the absolute class counts, determines the weighting. We will make this clarification explicit in the final version.

**CC3-ADDITIONAL BASELINES:** In addition to the existing six baselines (FedAvg, FedProx, PoC, FedEMD, S-FedAvg, GreedyFed), we have also included an additional baseline HiCS-FL [R1] as a recent client selection method that explicitly addresses data heterogeneity. This ensures that our evaluation covers both foundational methods and the latest state-of-the-art approaches.

[R1]: Chen, Huancheng, and Haris Vikalo. "Heterogeneity-guided client sampling: Towards fast and efficient non-iid federated learning." Advances in Neural Information Processing Systems (NeurIPS 2024).

---

> ### Author Response · Authors · 2025-10-03
> **Response to Common Comments of Reviewers**
>
> |  Table A  | **Setting: Mavericks + Dir(10)** |            |            |
> |:--------:|:--------------------------------:|:----------:|:----------:|
> |          | **OrganAMNIST**                  | **PathMNIST**             | **BloodMNIST**            |
> | FedAVG   | 0.65 ± 0.01 (0.18 ± 0.02)        | 0.51 ± 0.03 (0.01 ± 0.02) | 0.47 ± 0.12 (0.21 ± 0.26) |
> | FedProx  | 0.57 ± 0.01 (0.07 ± 0.05)        | 0.48 ± 0.04 (0.00 ± 0.00) | 0.48 ± 0.08 (0.21 ± 0.23) |
> | POC      | 0.66 ± 0.02 (0.19 ± 0.09)        | 0.51 ± 0.02 (0.01 ± 0.01) | 0.50 ± 0.04 (0.13 ± 0.22) |
> | FedEMD   | 0.67 ± 0.01 (0.25 ± 0.09)        | 0.55 ± 0.02 (0.07 ± 0.06) | 0.51 ± 0.03 (0.00 ± 0.00) |
> | GreedyFed| 0.65 ± 0.03 (0.20 ± 0.22)        | 0.50 ± 0.02 (0.00 ± 0.00) | 0.50 ± 0.01 (0.00 ± 0.00) |
> | S-FedAVG | 0.68 ± 0.01 (0.29 ± 0.06)        | 0.50 ± 0.01 (0.02 ± 0.04) | 0.51 ± 0.09 (0.04 ± 0.07) |
> | HiCS-FL  | 0.66 ± 0.02 (0.38 ± 0.05)        | 0.59 ± 0.01 (0.25 ± 0.11) | 0.43 ± 0.06 (0.17 ± 0.29) |
> | **FedMS (ours)** | 0.66 ± 0.04 (**0.69 ± 0.03**) | **0.66 ± 0.06** (**0.68 ± 0.26**) | 0.50 ± 0.01 (**0.35 ± 0.03**) |
>
>
> |  Table B  | **Setting: Mavericks + Dir(1)**  |            |            |
> |:--------:|:--------------------------------:|:----------:|:----------:|
> |          | **OrganAMNIST**                  | **PathMNIST**             | **BloodMNIST**            |
> | FedAVG   | 0.62 ± 0.05 (0.17 ± 0.06)        | 0.47 ± 0.06 (0.00 ± 0.00) | 0.55 ± 0.03 (0.21 ± 0.22) |
> | FedProx  | 0.54 ± 0.05 (0.09 ± 0.08)        | 0.38 ± 0.13 (0.00 ± 0.00) | 0.51 ± 0.03 (0.05 ± 0.07) |
> | POC      | 0.63 ± 0.03 (0.15 ± 0.11)        | 0.52 ± 0.01 (0.02 ± 0.02) | 0.42 ± 0.10 (0.12 ± 0.17) |
> | FedEMD   | 0.66 ± 0.01 (0.30 ± 0.10)        | 0.51 ± 0.04 (0.04 ± 0.06) | 0.54 ± 0.04 (0.12 ± 0.14) |
> | GreedyFed| 0.61 ± 0.01 (0.00 ± 0.00)        | 0.51 ± 0.03 (0.00 ± 0.00) | 0.52 ± 0.00 (0.00 ± 0.00) |
> | S-FedAVG | 0.64 ± 0.04 (0.19 ± 0.07)        | 0.50 ± 0.03 (0.06 ± 0.09) | 0.50 ± 0.06 (0.34 ± 0.35) |
> | HiCS-FL  | 0.66 ± 0.03 (0.38 ± 0.12)        | 0.57 ± 0.04 (0.30 ± 0.20) | 0.46 ± 0.03 (0.00 ± 0.00) |
> | **FedMS (ours)** | **0.67 ± 0.02** (**0.54 ± 0.07**) | **0.64 ± 0.03** (**0.72 ± 0.14**) | 0.49 ± 0.02 (**0.41 ± 0.41**) |
>
>
> |  Table C  | **Setting: Mavericks + Dir(0.1)** |            |            |
> |:--------:|:---------------------------------:|:----------:|:----------:|
> |          | **OrganAMNIST**                   | **PathMNIST**             | **BloodMNIST**            |
> | FedAVG   | 0.50 ± 0.06 (0.27 ± 0.02)         | 0.38 ± 0.09 (0.07 ± 0.08) | 0.36 ± 0.13 (0.17 ± 0.29) |
> | FedProx  | 0.44 ± 0.05 (0.27 ± 0.02)         | 0.36 ± 0.10 (0.09 ± 0.15) | 0.31 ± 0.05 (0.16 ± 0.29) |
> | POC      | 0.42 ± 0.14 (0.16 ± 0.15)         | 0.40 ± 0.08 (0.04 ± 0.05) | 0.32 ± 0.03 (0.04 ± 0.04) |
> | FedEMD   | 0.50 ± 0.09 (0.22 ± 0.20)         | 0.48 ± 0.05 (0.13 ± 0.13) | 0.40 ± 0.04 (0.09 ± 0.16) |
> | GreedyFed| 0.55 ± 0.03 (0.33 ± 0.12)         | 0.41 ± 0.05 (0.00 ± 0.00) | 0.37 ± 0.09 (0.00 ± 0.00) |
> | S-FedAVG | 0.52 ± 0.00 (0.27 ± 0.04)         | 0.50 ± 0.04 (0.14 ± 0.07) | 0.48 ± 0.03 (0.22 ± 0.22) |
> | HiCS-FL  | 0.45 ± 0.07 (0.10 ± 0.08)         | 0.43 ± 0.01 (0.07 ± 0.07) | 0.31 ± 0.09 (0.21 ± 0.36) |
> | **FedMS (ours)** | **0.62 ± 0.02** (**0.39 ± 0.25**)  | **0.60 ± 0.12** (**0.48 ± 0.42**) | **0.48 ± 0.11** (**0.55 ± 0.31**) |

---

### Decision · Action_Editor_YRHX · 2025-10-27

**Recommendation:** Accept with minor revision

**Audience:**

Yes

**Audience Explanation:**

The reviewers found the paper is relevant and interesting for the community. Originally the reviewers raised some concerns over the empirical validation. However, the authors then implemented additional validation using MedMNIST and this persuaded of the value of the proposed mechanism. However, the reviewers strong recommendation is to ensure that the assumptions are stated properly and the new experiments are properly referenced in the final version.

**Claims And Evidence:**

Yes

**Claims Explanation:**

The paper introduces a modified Shapley value (SV) computation for Federated Learning, focusing on a data distribution (in client-side) with rare classes.  Their techniques addresses the issue that standard uniform validation weighting systematically undervalues "Maverick" clients holding rare or underrepresented data. The proposed "Maverick-Shapley" framework utilizes a weighted utility function, based on metrics like per-class accuracy or recall, to assign greater importance to difficult classes where the model exhibits poor performance. This refined contribution score is then leveraged by FedMS, a client selection mechanism demonstrated on benchmark datasets to improve model generalization.